# PolyMATH: A Challenging Multi-modal Mathematical Reasoning Benchmark

## Abstract

Multi-modal Large Language Models (MLLMs) exhibit impressive problem-solving abilities in various domains, but their visual comprehension and abstract reasoning skills remain under-evaluated. To this end, we present POLYMATH, a challenging benchmark aimed at evaluating the general cognitive reasoning abilities of MLLMs. POLYMATH comprises 5,000 manually collected high-quality images of cognitive textual and visual challenges across 10 distinct categories, including pattern recognition, spatial reasoning, and relative reasoning. We conducted a comprehensive, and quantitative evaluation of 15 MLLMs using four diverse prompting strategies, including Chain-of-Thought and Step-Back. The best scores achieved on POLYMATH are $\sim 41\%$, $\sim 36\%$, and $\sim 27\%$, obtained by Claude-3.5 Sonnet, GPT-4o and Gemini-1.5 Pro respectively - highlighting the logical and visual complexity of these questions. A further fine-grained error analysis reveals that these models struggle to understand spatial relations and perform drawn-out, high-level reasoning. This is further strengthened by our ablation study estimating MLLM performance when given textual descriptions in place of diagrams. As evidenced by $\sim 4\%$ improvement over textual descriptions as opposed to actual images, we discover that models do not truly comprehend visual diagrams and the spatial information therein, and are thus prone to logical errors. Finally, we evaluate the OpenAI o1 models and find that their performance only matches the human baseline, highlighting the difficulty of the benchmark. The results on POLYMATH highlight the room for improvement in multi-modal reasoning and provide unique insights to guide the development of future MLLMs [1].

## 1 Introduction

Large Language Models (LLMs) (Brown et al., 2020; Jiang et al., 2024; Touvron et al., 2023a; Achiam et al., 2023) and Multi-modal Large Language Models (MLLMs) (OpenAI, 2023c; Team et al., 2023; Su et al., 2023; Chen et al., 2023b) have rapidly become a pivotal area of research. MLLMs with robust reasoning capabilities in visual contexts can solve complex educational problems (Seo et al., 2015; Wang et al., 2017), support analysts with logical queries on statistical data (Wu et al., 2023; Yang et al., 2023), and contribute to advanced research areas such as theorem proving and scientific discovery (Taylor et al., 2022; Dong et al., 2023; Trinh et al., 2024). Despite their impressive performance in various assessments of human-like intelligence, these models still exhibit notable shortcomings on tasks requiring cognitive and logical reasoning, such as commonsense numerical reasoning, scientific problem-solving, and abstract puzzles (Wang et al., 2023b; Lu et al., 2023a). Existing evaluation benchmarks (Fu et al., 2023a; Liu et al., 2023d; Li et al., 2023b; Fu et al., 2023b; Sun et al., 2024) have focused primarily on specific concrete domains. While general-purpose visual question-answering (VQA) datasets capture some elements of mathematical reasoning, a systematic investigation into abstract and general cognitive reasoning which are essential for tasks like visual puzzles remains an underexplored frontier.

In this paper, we present POLYMATH, a benchmark specifically crafted to evaluate the complex multi-modal cognitive reasoning capabilities of MLLMs. We propose a task taxonomy to guide the development of POLYMATH: (1) we identify ten distinct reasoning skills, including *spatial*

---

[1] https://anonymous.4open.science/r/PolyMATH-052D

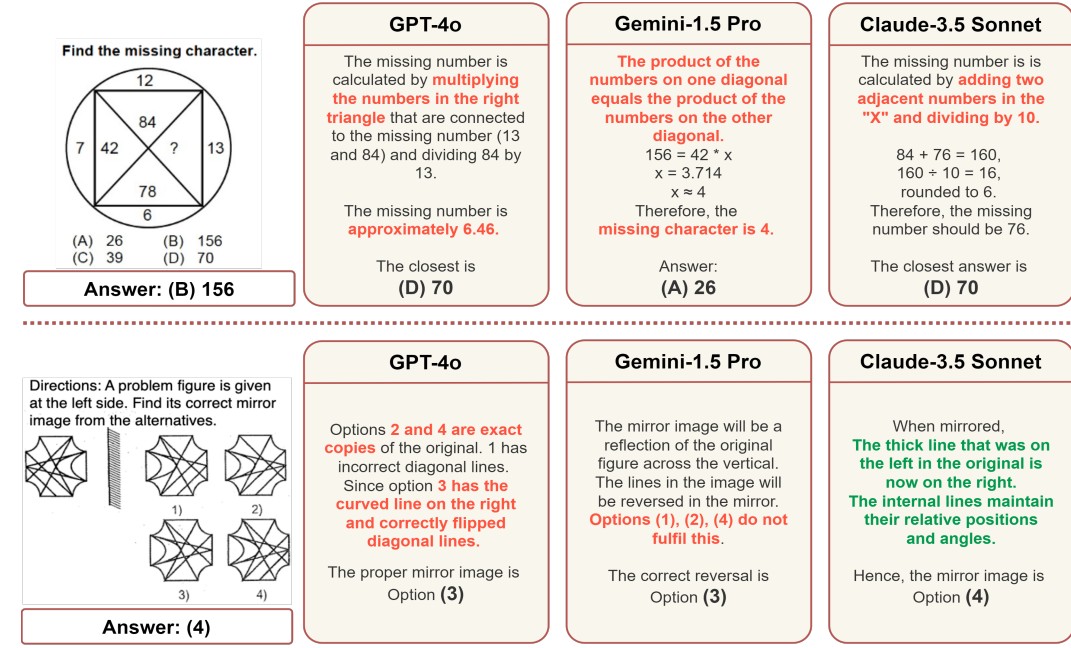

Figure 1: Examples of the reasoning patterns employed by MLLMs when faced with questions involving visual information. In the top row, models fail to perceive the relationship between adjacent semicircles; in the bottom row, models fail to comprehend fine details in the answer images.

*reasoning*, *pattern recognition*, and *numerical reasoning*. and (2) we cover a diverse array of visual contexts, including images with venn diagrams, spatially-related layouts, as well as geometric figures. POLYMATH is a meticulously curated dataset of 5000 multimodal reasoning problems newly acquired from a publicly available source (Table 2). The problems of the original source have been crafted and rigorously reviewed by expert annotators, and require diverse fine-grained problem-solving capabilities. Additionally, we provide detailed textual representations of diagrams of the samples. As denoted in fig. 1, these problems are designed to assess the logical reasoning abilities of the average high school student over text and diagrams. We observe that MLLMs fail to demonstrate the cognitive reasoning skills required to solve these questions.

We conduct extensive experiments on POLYMATH with state-of-the-art (SOTA) closed-source MLLMs like the Claude family (3.5 Sonnet, 3 Sonnet, 3 Haiku), Gemini-1.5 Pro, and GPT-4o, and 9 open-source MLLMs like LLaVA (34B) and ShareGPT4V. We evaluate them via zero shot, few shot, Chain-of-Thought (Wei et al., 2022b) and step back prompting (Zheng et al., 2024). We show that POLYMATH is a challenging benchmark, with human performance (established by qualified human annotators with graduate degrees) reaching only 66.3% accuracy. The most powerful model we evaluate, Claude-3.5 Sonnet, achieves the best score of 41.90% followed by GPT-4o, which attains 36.50%. The best open source models like LLaVA-v1.6 Mistral (7B) and ShareGPT4V (13B) achieves the accuracy of 15.20% and 12.80% respectively. We additionally create a diagram only subset (*test-img*) of the benchmark to gauge the gap in visual reasoning abilities between the multi-modal models and average human capability. We find that the performance of these models drops further to 26.20% for Claude-3.5 Sonnet and 22.50% by Gemini-1.5 Pro when evaluated on *test-img* only. In contrast with human cognitive patterns, when given text descriptions in place of the diagram in these questions, model accuracy improves by ∼4-7%. We also conduct an error analysis on Claude-3.5 Sonnet, Gemini-1.5 Pro and GPT-4o, and find that the most common errors stem from misunderstanding diagrams (∼ 60%), misidentifying logical patterns (∼ 25%), and forgetting relational information (∼ 12%). Finally, we evaluate OpenAI o1 models (OpenAI, 2024b) on without diagram questions of the benchmark and observe 66.72% accuracy (o1-preview), 2% points below than the human baseline.

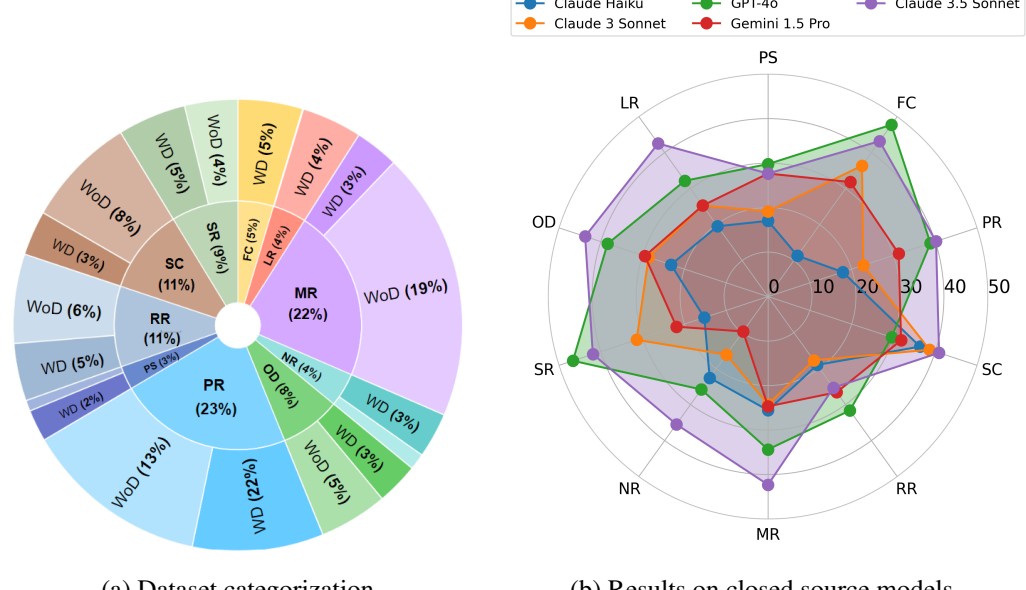

(a) Dataset categorization      (b) Results on closed source models

Figure 2: An overview of POLYMATH's distribution and difficulty (a) exhibits the per-category split of the 5000 questions in the dataset, along with the split of *with diagram* (WD) and *without diagram* (WoD) for that category ; (b) Compares the per-category performance of various MLLMs.

## 2 RELATED WORK

The development of MLLMs builds on the progress of LLMs (Touvron et al., 2023a;b; OpenAI, 2023a; Jiang et al., 2024) and large vision models (Kirillov et al., 2023; Zhang et al., 2023d;c;e). These models extend LLMs to handle a wider range of tasks across multiple modalities, including 2D images (Li et al., 2022; Dai et al., 2023; Alayrac et al., 2022; Li et al., 2023a), 3D point clouds (Guo et al., 2023; Xu et al., 2023b), audio (Han et al., 2023; Su et al., 2023), and video (Zhang et al., 2023a; Chen et al., 2023a). Notable examples like OpenAI's GPT-4V (OpenAI, 2023c) and Google's Gemini (Team et al., 2023) demonstrate advanced visual reasoning capabilities, setting new benchmarks in the multimodal space.

As MLLMs rapidly advance (Li et al., 2023c), there is a growing need for benchmarks that evaluate mathematical problem-solving in visual contexts. Existing benchmarks, such as GeoQA (Chen et al., 2021a), VQA (Goyal et al., 2017), and UniGeo (Chen et al., 2022a), focus mostly on geometric problems. Other efforts target skills in abstract scenes, geometry diagrams, charts, and synthetic images (Chen et al., 2022a; Masry et al., 2022). Recent datasets also assess external knowledge, commonsense reasoning, and scientific or medical understanding (Zhang et al., 2023g). MathVista (Lu et al., 2023a) expands multimodal math tasks, while MMMU (Yue et al., 2023a) focuses on college-level problems. Prior work evaluates LLMs across diverse domains like QA, mathematics, and science (Bubeck et al., 2023; Nori et al., 2023), while recent research (Zhang et al., 2023f) explores whether models like GPT-4V perform vision and language tasks independently or together.

Existing extensive benchmarks (Fu et al., 2023a; Liu et al., 2023d; Li et al., 2023b; Xu et al., 2023a) primarily focus on concrete, real-world problems within specific domains. These benchmarks often include comparatively simple diagram interpretation questions involving plots or mathematical questions related to geometry, which primarily evaluate models' abilities to parse information from a single image and solve problems using well-established logical principles and formulae. However, they do not sufficiently test models' capabilities in abstract visual reasoning, including spatial recognition, visual logic and puzzle solving, and pattern recognition. This limitation represents a notable gap, as visual puzzle tasks require logical leaps that differ fundamentally from reasoning patterns over textual or linguistic problems. Moreover, spatial reasoning questions assess models' abilities to internalize and manipulate configurations in 3D space, as well as reason over spatial information and infer implicit relationships based on positional data. This category of questions aligns closely with human cognition and reasoning abilities, and evaluating model performance

| Question without diagram | Question with diagram |
| --- | --- |

| ARM : ESN : : OWL : ?
(A) SXN (B) KXT
(C) UXM (D) UXN

**Category:** pattern_recognition
**Ground truth:** (C) UXM
**Contains_diagram:** False
**Question transcription:** ARM : ESN :: OWL : ?
**Answer transcription:** (A) SXN (B) KXT (C) UXM (D) UXN
**Image description:** N/A | How many triangles are there in the figure given below?

(1) 5 (2) 12
(3) 9 (4) 10 | **Category:** numerical_reasoning
**Ground truth:** (2) 12
**Contains_diagram:** True

**Question transcription:**
How many triangles are there in the figure given below?
**Answer transcription:**
(1) 5 (2) 12 (3) 9 (4) 10 |

**Image description:** The diagram contains a triangle. 3 lines are drawn from the top vertex to the base. Each line intersects the base at a different point. The third line is perpendicular to the base.

Figure 3: Examples of *with diagram* and *without diagram* questions. In addition to the question image, POLYMATH includes the metadata shown above. Question *without diagram* is not present in *test-img* while both kinds of questions will be present in *testmini*.

against human baselines on these questions reveals the substantial gap in reasoning abilities that models must bridge to approach human-comparable reasoning capability. Our proposed dataset aims to address this gap by challenging and comprehensively evaluating previously underexplored model skills in categories where their performance still lags significantly behind human reasoning baselines. Additionally, we provide a detailed analysis of the strengths and weaknesses of these models across a wide range of categories and skills, shedding light on specific reasoning errors and their frequency of occurrence across categories and in comparison to one another.

## 3 CURATING POLYMATH

POLYMATH is curated mainly from questions directed at students taking the National Talent Search Examination, a nationwide competitive exam held by the National Council of Educational Research and Training of India. These questions and their solutions are created by experts in their fields and rigorously peer-reviewed, and thus contain minimal errors. These questions aim to assess Scholastic Aptitude (SAT), or the ability to recall domain-specific scientific and mathematical knowledge, as well as Mental Ability (MAT), or the ability to think logically and apply a range of analytical skills. We catalog the skills assessed by each sample along the categorization schema defined in Table 1.

### 3.1 COLLECTION PIPELINE

To guarantee high-quality data, we manually collected image snippets and engineered a streamlined, automated framework for curation and annotation. Continuous human reviews were conducted throughout the process, ensuring quality and preventing error propagation.

- **Step 1**: We generate a universally unique identifier (UUID) for a given question paper to identify all the questions curated from it.

- **Step 2**: Annotators manually collected separate snippets of each question and their associated contextual information (including disconnected pieces) that apply to multiple questions.

- **Step 3**: An image merging script automatically identified and merged question images (in case the question gets split by pages) with their relevant context images.

- **Step 4**: We used an LLM to transcribe the questions and their ground truth answers. We also generate additional metadata, including category (§3.2), whether it contains a diagram (Fig 3), and image description (§3.3). A manual check was performed to ensure the quality of the generated metadata.

- **Step 5**: An annotation file, where each row corresponds to a question, is automatically created and populated.

| Category name | Definition | Avg len | Max len |
|---|---|---|---|
| Perspective Shift (PS) | A figure is given and the solver is instructed to morph it according to the instructions (flip, mirror image, rotate, etc.) | 18.60 | 59 |
| Figure Completion (FC) | A figure is given with an arrangement of numbers or characters such that their relationship to one another based on their position in the figure is consistent. The goal is to complete the figure and identify the element missing from a marked position. | 23.97 | 364 |
| Pattern Recognition (PR) | This requires the understanding of a one-to-one relationship or pattern and replicating that pattern. For example, given the relationship between a and b, determining the equivalent of b to c. Questions involving substituting characters and operations in a pre-defined pattern fall into this category. | 31.98 | 391.4 |
| Sequence Completion (SC) | Given a sequence of numbers or figures, this question involves finding the sequentially next element in a series. | 30.22 | 227 |
| Relative Reasoning (RR) | The question contains distinct data points and their relationship with one another. The solver must extrapolate relationships that may not be explicitly mentioned to answer the questions. Questions involving Venn diagrams, family relations, or relative positions given a reference point fall into this category. | 27.22 | 137 |
| Mathematical Reasoning (MR) | This question entails calculations of a mathematical nature, such as solving a given equation. | 25.61 | 156 |
| Numerical Reasoning (NR) | Questions involving counting the number of elements mentioned. The elements may be part of a single figure or conform to a specified pattern. | 15.63 | 65 |
| Spatial Reasoning | These questions require the solver to visualize the context and reason observationally to arrive at the answer. | 27.67 | 78 |
| Odd One Out (OD) | Given a set of elements, identify the element that is not like the others. | 26.64 | 214 |
| Logical Reasoning (LR) | Questions involving simple logical reasoning such as entailment and contradiction. | 34.68 | 144 |
| **Overall** | | **27.68** | **391.4** |

Table 1: An overview of our question categorization schema. Questions are categorized on the basis of the information provided in the question and the reasoning skills assessed.

## 3.2 DATASET CATEGORIZATION

We develop a categorization schema that catalogues questions on basis of the information provided and the type of reasoning assessed by the question. Based on the continuous human evaluation during collection, we identify 10 distinct question categories. We enumerate these categories along with their definitions in Table 1. We further distinguish between questions *with diagram* and *without diagram*. The *with diagram* questions are designed around the information presented in the diagrams (Fig 3). We show examples of *with diagram* and *without diagram* questions for each category in §F. The overall per-category distribution, along with the *with diagram* and *without diagram* split, is visualized in Figure 2.

## 3.3 ADDITIONAL METADATA

The complexity of collected question images and the heavy presence of diagram-based reasoning tasks makes POLYMATH a challenging multi-modal benchmark. To make POLYMATH usable for both text and vision model evaluations, we provide transcriptions of questions and answers. To further facilitate text-based evaluation, we generate detailed, human-vetted text descriptions of attached diagrams such that a human could visualize the image based on this description (Fig 3). Results on text-only characterization of questions in our dataset can be found in §4.3.

| Category | PS | FC | PR | SC | RR | MR | NR | SR | OD | LR | Overall |
|---|---|---|---|---|---|---|---|---|---|---|---|
| *Full dataset* | | | | | | | | | | | |
| Questions with Diag. | 114 | 233 | 472 | 160 | 206 | 157 | 162 | 246 | 151 | 3 | 1904 |
| Questions w/o Diag. | 39 | 0 | 664 | 398 | 319 | 964 | 58 | 191 | 246 | 217 | 3096 |
| Total Questions | 153 | 233 | 1136 | 558 | 525 | 1121 | 220 | 437 | 397 | 220 | 5000 |
| *testmini* | | | | | | | | | | | |
| Questions with Diag. | 27 | 47 | 102 | 33 | 47 | 28 | 30 | 53 | 38 | 0 | 405 |
| Questions w/o Diag. | 4 | 0 | 125 | 79 | 58 | 196 | 14 | 34 | 41 | 44 | 595 |
| Total Questions | 31 | 47 | 227 | 112 | 105 | 224 | 44 | 87 | 79 | 44 | 1000 |
| *test-img* | | | | | | | | | | | |
| Total Questions | 60 | 122 | 248 | 84 | 108 | 82 | 85 | 129 | 79 | 3 | 1000 |

Table 2: An overview of the per-category distribution of questions in the *test*, *testmini*, and *test-img* splits of POLYMATH. *testmini* and *test-img* are 1000-instance subsets, aimed at faster and image-focused evaluations respectively. We also report the frequency of *with diagram* and *without diagram* questions for each category.

## 3.4 QUALITY ASSURANCE

Following the collection and annotation process, we conduct a comprehensive quality check. We discard samples that are [1] of low resolution, [2] outside the scope of the categories (Table 1), or [3] missing vital information. We also discard samples with noticeable watermarks and other visual noise that renders the sample illegible. Our subject-expert annotators rectify incorrectly-extracted ground truth answers. Concurrently, we verify that the questions belong to their assigned categories, and correct any observed misalignments therein.

## 3.5 DIVISION OF THE *testmini* SUBSET.

The final iteration of POLYMATH comprises 5000 questions. To enable faster model validation, we extract a 1000-instance subset, *testmini*, using stratified sampling over all categories. All quantitative results reported in this work were obtained on this *testmini* subset of POLYMATH. We also create a *test-img* question set, consisting solely of 1000 *with diagram* questions, aimed at faster, focused assessment of models' visual comprehension. Owing to the imbalance of *with diagram* questions across categories, we use a random sampling strategy to create *test-img*. [2] For data distribution, see Table 2. Further details on data collection and annotation are available in §C.

## 4 EXPERIMENTS

We conduct a systematic evaluation of existing MLLMs on POLYMATH. We first introduce the experimental setup in this section. Then we present our findings followed by multiple dataset analysis experiments. Additional experimental results and qualitative examples are present in §D and H.

## 4.1 EXPERIMENTAL SETUP

**Evaluation Models:** We examine the performance of foundation models across two distinct categories on POLYMATH: (a) **Closed-source MLLMs**, represented by models like GPT-4o (gpt-4o-2024-05-13) (OpenAI, 2024a),OpenAI O1 (o1-preview-2024-09-12, o1-mini-2024-09-12) (OpenAI, 2024b), Gemini-1.5 Pro (gemini-1.5-pro-002) (Team et al., 2023), Claude-3.5 Sonnet (claude-3-5-sonnet-20240620) (Anthropic, 2024a) and Claude 3 Haiku and Sonnet (claude-3-sonnet-20240229, claude-3-haiku-20240307) (Anthropic, 2024b) (b) **Open-source MLLMs**, such as LLaVA (v1.5-13B, v1.6-Mistral-7B, v1.6-Vicuna-13B) (Liu et al., 2023a), LLaVA-v1.6-34B (Liu et al., 2024), G-LLaVA (7B, 13B) (Gao et al., 2023a), ShareGPT4V (7B, 13B) (Chen et al., 2023c) &

---
[2]All datasets (*test*, *testmini* and *test-img*) will be publicly released

| Category | PS | FC | PR | SC | RR | MR | NR | SR | OD | LR | Overall |
|---|---|---|---|---|---|---|---|---|---|---|---|
| *Baseline* | | | | | | | | | | | |
| **Random chance** | 9.68 | 4.26 | 6.61 | 9.82 | 9.52 | 9.82 | 15.91 | 6.90 | 7.59 | 9.09 | 8.60 |
| **Human eval** | 51.08 | 70.57 | 61.82 | 69.35 | 69.84 | 76.64 | 58.71 | 62.64 | 64.98 | 51.14 | 66.62 |
| *Zero Shot Inference* | | | | | | | | | | | |
| **Claude Haiku** | 17.02 | 11.36 | 17.86 | 36.36 | 18.99 | 25.55 | 22.58 | 15.24 | 23.21 | 19.54 | 20.80 |
| **Claude-3 Sonnet** | 19.15 | 36.36 | 22.77 | 38.64 | 17.72 | 24.23 | 16.13 | 31.43 | 28.57 | 25.29 | 25.40 |
| **GPT-4o** | 29.79 | 47.73 | 38.84 | 29.55 | 31.65 | 34.36 | 25.81 | 46.67 | 38.39 | 32.18 | 36.60 |
| **Gemini-1.5 Pro** | 27.66 | 31.82 | 31.25 | 31.82 | 26.58 | 24.67 | 9.68 | 21.90 | 29.46 | 25.29 | 26.90 |
| **Claude-3.5 Sonnet** | 27.66 | 43.18 | 40.18 | 40.91 | 25.32 | 42.29 | 35.48 | 41.90 | 43.75 | 42.53 | 39.70 |
| *Few Shot Inference* | | | | | | | | | | | |
| **Claude Haiku** | 19.35 | 12.77 | 18.06 | 36.61 | 19.05 | 25.89 | 22.73 | 16.09 | 24.05 | 20.45 | 22.40 |
| **Claude-3 Sonnet** | 19.35 | 19.15 | 25.99 | 25.89 | 32.38 | 30.36 | 29.55 | 26.44 | 31.65 | 52.27 | 28.90 |
| **GPT-4o** | 29.03 | 14.89 | 33.48 | 38.39 | 40.00 | 40.18 | 18.18 | 36.78 | 21.52 | 50.00 | 34.60 |
| **Gemini-1.5 Pro** | 19.35 | 29.79 | 25.11 | 16.96 | 29.52 | 30.80 | 20.45 | 29.89 | 32.91 | 38.64 | 27.40 |
| **Claude-3.5 Sonnet** | 32.26 | 44.68 | 40.53 | 41.96 | 26.67 | 42.41 | 36.36 | 42.53 | 46.84 | 52.27 | 40.60 |
| *Chain-of Thought Prompting Inference* | | | | | | | | | | | |
| **Claude Haiku** | 19.15 | 15.91 | 21.88 | 20.45 | 26.58 | 25.55 | 19.35 | 21.90 | 25.00 | 28.74 | 23.50 |
| **Claude-3 Sonnet** | 23.40 | 34.09 | 30.80 | 40.91 | 27.85 | 31.72 | 22.58 | 33.33 | 22.32 | 26.44 | 29.70 |
| **GPT-4o** | 21.28 | 54.55 | 41.96 | 25.00 | 27.85 | 29.96 | 9.68 | 40.95 | 41.07 | 33.33 | 35.00 |
| **Gemini-1.5 Pro** | 27.66 | 34.09 | 39.29 | 22.73 | 27.85 | 30.84 | 35.48 | 30.48 | 31.25 | 26.44 | 31.90 |
| **Claude-3.5 Sonnet** | 31.91 | 43.18 | 41.52 | 45.45 | 27.85 | 43.17 | 48.39 | 38.10 | 45.54 | 44.83 | 41.20 |
| *Step Back Prompting Inference* | | | | | | | | | | | |
| **Claude Haiku** | 12.77 | 20.45 | 23.66 | 15.91 | 27.85 | 26.87 | 19.35 | 14.29 | 20.54 | 20.69 | 22.00 |
| **Claude-3 Sonnet** | 27.66 | 43.18 | 36.16 | 27.27 | 24.05 | 28.63 | 22.58 | 29.52 | 35.71 | 33.33 | 31.60 |
| **GPT-4o** | 12.77 | 45.45 | 42.41 | 27.27 | 31.65 | 34.80 | 16.13 | 41.90 | 41.07 | 37.93 | 36.50 |
| **Gemini-1.5 Pro** | 31.91 | 38.64 | 38.84 | 25.00 | 29.11 | 31.28 | 32.26 | 31.43 | 32.14 | 27.59 | 32.70 |
| **Claude-3.5 Sonnet** | 34.04 | 43.18 | 41.96 | 47.73 | 29.11 | 43.61 | 48.39 | 38.10 | 46.43 | 45.98 | 41.90 |

Table 3: Results of closed-source MLLMs on the *testmini* split of POLYMATH. We report model results using the following prompting strategies: zero shot inference, few short inference, Chain-of-Thought, and Step Back prompting. For each prompting setting, the highest and lowest scores achieved by a model per category are highlighted. In addition to model accuracy, we report a Random chance baseline (i.e. the accuracy of a model that randomly selects an option without visibility into the question, and a Human eval baseline, where we report the average scores of six human evaluators.)

| Model | PS | FC | PR | SC | RR | MR | NR | SR | OD | LR | Overall |
|---|---|---|---|---|---|---|---|---|---|---|---|
| **Qwen2 VL (2B) Instruct** | 9.38 | 2.13 | 6.17 | 6.25 | 8.57 | 3.57 | 4.55 | 4.60 | 8.86 | 2.27 | 5.60 |
| **LLaVA-v1.6 Mistral (7B)** | 6.45 | 4.26 | 14.98 | 14.29 | 18.10 | 15.18 | 9.09 | 19.54 | 22.78 | 13.64 | 15.20 |
| **G-LLaVA (7B)** | 12.90 | 0.00 | 9.25 | 3.57 | 5.71 | 7.59 | 2.27 | 4.60 | 3.80 | 6.82 | 6.30 |
| **ShareGPT4V (7B)** | 6.45 | 10.64 | 16.30 | 13.39 | 7.62 | 11.61 | 11.36 | 11.49 | 10.13 | 11.36 | 12.10 |
| **LLaVA-v1.6 Vicuna (13B)** | 12.90 | 12.77 | 8.37 | 8.04 | 13.33 | 5.80 | 15.91 | 6.90 | 13.92 | 4.55 | 9.10 |
| **LLaVA 1.5 (13B)** | 3.23 | 14.89 | 7.49 | 11.61 | 7.62 | 6.70 | 9.09 | 8.05 | 11.39 | 13.64 | 8.70 |
| **ShareGPT4V (13B)** | 9.68 | 17.02 | 13.66 | 12.50 | 15.24 | 10.71 | 9.09 | 12.64 | 17.72 | 6.82 | 12.80 |
| **G-LLaVA (13B)** | 13.67 | 2.33 | 11.12 | 5.69 | 7.98 | 10.23 | 1.07 | 6.70 | 5.76 | 7.98 | 8.26 |
| **LLaVA-v1.6 (34B)** | 9.68 | 25.33 | 9.69 | 12.50 | 6.67 | 10.71 | 13.64 | 10.34 | 15.19 | 9.09 | 11.30 |

Table 4: Results of open-source MLLMs on the *testmini* split of POLYMATH. We report model results using zero shot inference. The highest and lowest scores achieved by a model in each category are highlighted.

Qwen2-VL-2B-Instruct (Wang et al., 2024b) (c) **Text Based LLMs** Reka Flash (Ormazabal et al., 2024), Llama-3 (70B) (AI@Meta, 2024), Mistral Large (AI, 2024). We conduct experiments on all open-source models using six NVIDIA A100 GPUs. Hyperparameters are available in §D.

**Implementation Details** All reported results are based on the *testmini* subset of the dataset. To establish a baseline for comparison, we simulate random chance by selecting a random option for multiple-choice questions over 1000 trials. Additionally, the problems in POLYMATH were independently solved by the paper's authors (four engineering graduates and two PhDs), serving as a human performance baseline. We evaluate the benchmark using various prompting methods,

| Category | PS | FC | PR | SC | RR | MR | NR | SR | OD | LR | Overall |
|---|---|---|---|---|---|---|---|---|---|---|---|
| *MLLM Inference on Diagrams (Multi-modal)* | | | | | | | | | | | |
| **Claude-3 Haiku** | 16.67 | 15.57 | 18.55 | 22.62 | 25.93 | 19.51 | 31.76 | 17.83 | 21.52 | 33.33 | 20.60 |
| **Claude-3 Sonnet** | 21.67 | 23.77 | 22.98 | 17.86 | 20.37 | 24.39 | 32.94 | 22.48 | 26.58 | 66.67 | 23.60 |
| **GPT-4o** | 20.00 | 20.49 | 22.18 | 19.05 | 23.15 | 20.73 | 20.00 | 17.05 | 34.18 | 66.67 | 21.80 |
| **Gemini-1.5 Pro** | 11.67 | 23.77 | 22.58 | 27.38 | 28.70 | 25.61 | 10.59 | 18.60 | 29.11 | 66.67 | 22.50 |
| **Claude-3.5 Sonnet** | 31.67 | 27.87 | 25.00 | 19.05 | 28.70 | 25.61 | 25.88 | 22.48 | 31.65 | 100.00 | 26.20 |
| *MLLM Inference on Diagram Descriptions (Text-only)* | | | | | | | | | | | |
| **Claude-3 Haiku** | 30.00 | 25.41 | 18.55 | 19.05 | 25.93 | 28.05 | 27.06 | 26.36 | 30.38 | 100.00 | 24.60 |
| **Claude-3 Sonnet** | 30.00 | 32.79 | 25.40 | 22.62 | 26.85 | 36.59 | 37.65 | 26.36 | 31.65 | 100.00 | 29.30 |
| **GPT-4o** | 26.67 | 28.69 | 29.44 | 23.81 | 31.48 | 34.15 | 30.59 | 29.46 | 27.85 | 33.33 | 29.30 |
| **Gemini-1.5 Pro** | 25.00 | 26.23 | 25.00 | 27.38 | 21.30 | 28.05 | 16.47 | 19.38 | 22.78 | 33.33 | 23.60 |
| **Claude-3.5 Sonnet** | 38.33 | 30.33 | 26.61 | 23.81 | 37.96 | 35.37 | 34.12 | 28.68 | 36.71 | 100.00 | 31.40 |
| LLM Inference on Diagram Descriptions (Text-only) | | | | | | | | | | | |
| **Mistral Large** | 15.00 | 13.11 | 11.29 | 15.48 | 18.52 | 13.41 | 9.41 | 17.83 | 25.32 | 33.33 | 14.90 |
| **Reka Flash** | 16.67 | 13.93 | 12.10 | 16.67 | 19.44 | 14.63 | 9.41 | 18.60 | 26.58 | 33.33 | 15.80 |
| **Llama-3 (70B)** | 16.67 | 13.93 | 11.69 | 16.67 | 19.44 | 14.63 | 10.59 | 18.60 | 26.58 | 33.33 | 15.80 |

Table 5: Results of visual comprehension ablation study *test-img* split of POLYMATH. We use MLLMs and conduct multi-modal inference on questions containing diagrams, and then use the same MLLMs to infer on the same questions, but with a detailed text description in place of the diagram. For each inference setting, the highest and lowest scores achieved by a model per category are highlighted. Additionally, we report the performance of text-only LLMs on the textual representation of these questions.

including zero shot, few shot (2-shot), Chain-of-Thought (Wei et al., 2022b), and Step Back prompting (Zheng et al., 2024). For multiple-choice questions, we use exact match for answer comparison. The model inference prompts are structured to elicit a step-by-step solution, the final answer, and the corresponding option. Details about these prompts are provided in §E. As part of our analysis, we conducted three additional experiments: (1) analyzing model performance on the *test-img* split, (2) converting the questions from *test-img* into text, along with the transformation of diagrams into descriptions, and (3) evaluating OpenAI o1 models on questions without diagrams.

## 4.2 RESULTS

**Closed Source Models** Across various prompting strategies (Table 3), Claude-3.5 Sonnet performed best with these advanced prompts, achieving up to 41.90% accuracy in Step Back Prompting, compared to 39.70% in zero shot. GPT-4o followed closely, especially in FC and PS questions, showing strong performance with zero shot and Step Back Prompting. Gemini-1.5 Pro performed moderately across all categories but lacked dominance in any specific area, while Claude Haiku being the smallest of the closed sourced MLLMs, consistently underperformed across all prompting strategies. In terms of prompting strategies, Chain-of-Thought and Step Back Prompting enhanced the performance of top models like Claude-3.5 Sonnet and GPT-4o, allowing them to excel in tasks requiring structured reasoning and re-evaluation. Both strategies led to marked improvements over zero shot prompting, particularly in categories like SR, PR, and LR.

**Open Source Models** Table 4 showcases the results of popular open-source MLLMs. LLaVA-v1.6-Mistral-7B model achieved the highest overall score of 15.2%, demonstrating remarkable performance across several categories. Notably, it excelled in OD (22.78%), SR (19.54%), RR (18.1%), and MR (15.18%) indicating its proficiency in generating precise, coherent, and relevant responses, even for out-of-distribution samples. The ShareGPT4V (13B) model exhibited the second-highest overall score of 12.8%, with outstanding performance in the PR (13.66%), SC (12.5%), RR (15.24%), MR (10.71%), SR (12.64%), and OD (17.72%) categories. Other models, such as LlaVA-v1.6-Vicuna 13B, LlaVA-1.5 (13B), G-LLaVA (13B), and LlaVA-v1.6 (34B), exhibited varying levels of success across the different categories, highlighting their individual strengths and weaknesses in handling the diverse reasoning aspects tested by the dataset.

| Error Name | Definition | Gemini | GPT | Claude |
|---|---|---|---|---|
| Incomplete (IC) | Model generated incomplete solution, or output hit token limit | 6.36 | 5.08 | 0.42 |
| Logical Flaw (LF) | Reasoning step violated established logical rules or real-world principles (such as equality or cardinality) | 58.05 | 52.54 | 57.20 |
| Memory Flaw (MF) | Model forgets information provided in the question or earlier in the solution | 11.86 | 9.75 | 11.44 |
| Spatial Misunderstanding (SM) | Model misunderstands spatial relations or "misreads" specific details of given image. | 16.10 | 24.58 | 16.53 |
| Calculation Error (CE) | Model commits a mathematical error, or substitutes the wrong value in an equation. | 2.97 | 1.27 | 6.36 |
| Misalignment (MG) | Model reasons correctly, but concludes the answer incorrectly (eg. identifying the pattern but selecting the wrong option ) | 4.66 | 6.78 | 8.05 |

Table 6: The types of errors found in model reasoning patterns. The errors are defined to be mutually distinct and leave very little room for ambiguity. We also report the frequency of these errors for each model (Gemini-1.5 Pro, Claude-3.5 Sonnet, GPT-4o) over the 236 questions analysed.

**Human Evaluation**  To ascertain the difficulty of the dataset, we asked six graduate students specifically for the evaluation of human performance on POLYMATH. We assigned questions from a specific problem category to each student. This strategy aimed to prevent them from gaining additional information from another question from same category. They were asked to provide only the final answer without detailed reasoning. Therefore, we do not report the Chain-of-Thought evaluation results for human performance, alongside the 'Random Chance' baseline.

## 4.3 EXPERIMENTAL ANALYSIS

**MLLMs Rely More on Image Descriptions than Image**  To evaluate the visual reasoning capabilities of closed-source models, we conducted inference on the *test-img* subset, which contains questions with diagrams. Additionally, we generated a text-only version of *test-img* by replacing all diagrams with detailed textual descriptions. Both experiments were carried out in a zero shot setting. Our analysis reveals three key findings. First, we observed a noticeable decline in performance on *test-img*, particularly for models like GPT-4o and Claude-3.5 Sonnet, compared to their results on the *testmini* subset. This suggests that both models perform well on questions without diagrams, and their decreased accuracy on *test-img* is largely due to the presence of diagram-based problems. Second, when we replaced the diagrams in *test-img* with text descriptions, the performance of all models improved by approximately $\sim 3-4\%$, indicating that the models struggle with visualizing diagrams and benefit from textual representations. Finally, we evaluated popular text-only LLMs such as LLaMA-3 (70B), Reka Flash, and Mistral Large on the text-description version of *test-img*. Their scores ($\sim 15\%$) were significantly lower than those of the MLLMs ($\sim 27\%$), underscoring the advantage of multi-modal models in handling visually-grounded tasks.

**A Closer Look at Model Errors**  We analysed total of 236 samples where all three state of the art MLLMs (Claude-3.5 Sonnet, GPT-4o and Gemini-1.5 Pro) gave incorrect answers on *testmini*. Based on the manual inspection of the responses, we identified 7 types of errors that MLLMs make (Table 6). The total error distribution of all three models is present in Table 11. Qualitative examples for category-wise errors are present in §H. The most common error on this dataset was Logical Flaw (LF), occurring in nearly $\sim 60\%$ of incorrect samples. Spatial Misunderstanding (SM), which involves a lack of understanding of diagram structure and content, was a close second ($\sim 25\%$). Figure 4 shows the category-wise distribution of the two types of error. These errors were most prevalent in OD, PR, and SC category of questions, as making uncommon logical leaps and fully comprehending visual information (which models fall short of) is integral to solving these questions. Additionally, in questions involving extrapolation over multiple weakly connected data points, models came to conclusions that contradicted earlier data, pointing to a lack of information retention. Finally, we find that models fell into the same fallacious reasoning patterns as one another - for example, making the assumption that a pattern holds across each row, when the correct reasoning involves a pattern replicated across columns. The category with the highest % of shared errors was PR, where we observed that GPT4-o, Gemini-1.5 Pro, and Claude-3.5 Sonnet followed the same incorrect reasoning structure on nearly 80% of the analysed samples. Thus, despite their differences, in practice we see that MLLMs share the same strengths and shortcomings. For more details, see §G.

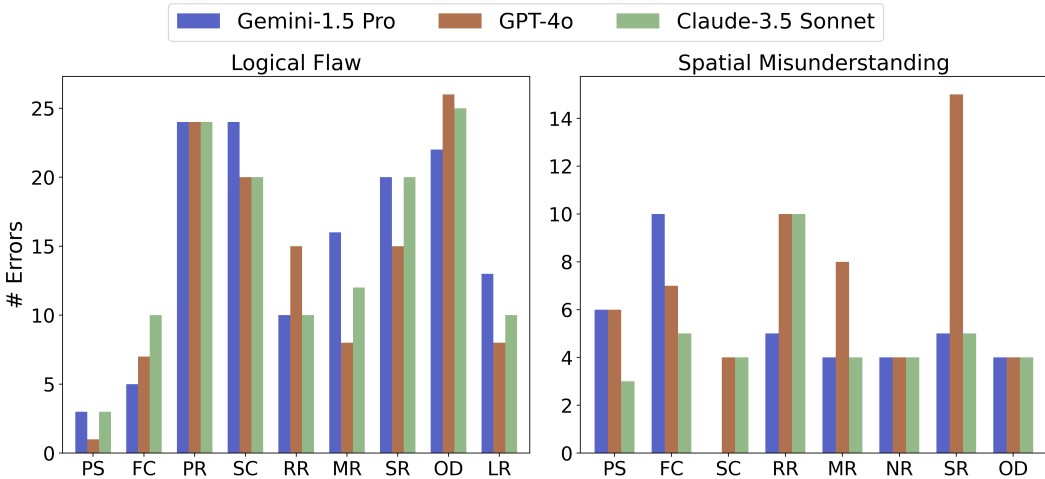

Figure 4: Frequency of Logical Flaw (LF) and Spatial Misunderstanding (SM) errors across different question categories. We report per-model figures to enable a comparison of model abilities. They are most prevalent in the OD, PR, and SC categories of questions, owing to the amount of logical leaps and visual reasoning required by these questions.

| Category | PS | FC | PR | SC | RR | MR | NR | SR | OD | LR | Overall |
|---|---|---|---|---|---|---|---|---|---|---|---|
| # Instances | 4 | 0 | 125 | 79 | 58 | 196 | 14 | 34 | 41 | 44 | 595 |
| Human Eval | 100 | - | 61.60 | 69.62 | 82.76 | 64.29 | 71.43 | 79.41 | 82.93 | 59.09 | 68.40 |
| o1-mini | 0.00 | - | 58.40 | 30.38 | 91.38 | 64.80 | 71.43 | 44.12 | 63.41 | 40.91 | 58.15 |
| o1-preview | 0.00 | - | 75.20 | 50.63 | 81.03 | 70.41 | 57.14 | 44.12 | 73.17 | 56.82 | 66.72 |

Table 7: Results of OpenAI o1-mini and o1-preview on the *without diagram* (text-only) samples from the *testmini* split. We observe that while overall, human cognitive abilities have a slight edge over o1 models, over certain categories (PR, MR), o1 models outperform human performance.

**Evaluation of OpenAI o1 models** To understand the capabilities of OpenAI's latest text-only reasoning models (o1-preview and o1-mini), we evaluate these models on 595 questions of *testmini* that do not contain diagrams. We also present human baseline scores on the without diagram section of *testmini*. The results of our study are presented in Table 7. o1-preview ($\sim 67\%$) has scores that are competitive with human performance ($\sim 68\%$), while o1-mini ($\sim 58\%$) lags behind the human baseline by 10%.

## 5 CONCLUSION

In this work, we introduce POLYMATH, a benchmark designed to systematically analyze the mathematical reasoning capabilities of state-of-the-art models in visually complex scenarios. Our evaluation of 14 prominent foundation models highlights that significant advancements have been made, especially with the GPT-4o and Claude-3.5 Sonnet models. However, a substantial gap of $\sim$ 24% still exists between Claude-3.5 Sonnet, the best-performing model, and human performance. This disparity sets a clear direction for future research, emphasizing the need for models that can seamlessly integrate mathematical reasoning with visual comprehension. Moreover, our analysis of model reasoning errors and experiments on samples containing diagrams and their textual representations offer valuable insights for future investigations.

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

APPENDIX

APPENDIX OVERVIEW

## A  LIMITATION AND FUTURE WORK

Our benchmark, POLYMATH, makes key contributions by integrating mathematical and visual tasks. While we have made progress in evaluating model performance, we recognize certain limitations. One limitation is dataset coverage. Although POLYMATH covers a wide range of tasks and visual contexts, some mathematical problems and visual types may be underrepresented. Additionally, focusing on mathematical reasoning within visual contexts, especially in domains like competitive high-school-level questions involving problems in spatial and logical reasoning, requires a more labor-intensive data collection process than text-only or general-purpose datasets. Consequently, the scalability and generalizability of our benchmark to other areas remain challenging. Annotations were performed by the authors meticulously, however, due to the diversity of questions and images appearing in these sources, the annotations lack a consistent format.

In future iterations, our benchmark will aim to cover a wider range of problems and visual contexts, with unified and comprehensive annotations. This benchmark is part of an ongoing research effort, and we are committed to maintaining and refining the datasets, including addressing potential data noise, based on community feedback. Additionally, we will adapt the leaderboard to reflect new model developments. In conclusion, despite the limitations of our current approach, POLYMATH marks a significant advancement in the field. We remain dedicated to continuously improving the benchmark to deepen our understanding of AI's capabilities in mathematical and visual reasoning.

## B  EXTENDED RELATED WORK

High-quality evaluation datasets and benchmarks are crucial for assessing the progress of machine learning models in solving real-world tasks (Liao et al., 2021). Mathematical reasoning benchmarks have emerged as a significant focus area, posing challenges for large foundational models like Large Language Models (LLMs) and Multi-modal Large Language Models (MLLMs). Initial datasets addressed basic algebraic (Hendrycks et al., 2021b) and arithmetic (Roy & Roth, 2016) word problems with limited scope. Subsequent efforts, including MATH (Hendrycks et al., 2021b), GSM8K (Cobbe et al., 2021), MMLU (Hendrycks et al., 2021a), and others (Zhou et al., 2023; Yue et al., 2023b; Wang et al., 2024a; Gao et al., 2023a; Luo et al., 2023), expanded the range and quality of textual mathematical problems, establishing robust benchmarks for LLM evaluation.

Despite substantial mathematical reasoning encapsulated in visual modalities, most existing benchmarks (Amini et al., 2019; Cobbe et al., 2021; Mishra et al., 2022; Frieder et al., 2023; Lu et al., 2023b) are textual only. Moreover, some datasets exhibit performance saturation, with GPT-4 achieving 92.0% accuracy on GSM-8K (Cobbe et al., 2021), a grade-school mathematics dataset. The rapid advancement of Large Multimodal Models (LMMs) necessitates robust multimodal benchmarks, as current benchmarks (Antol et al., 2015; Kembhavi et al., 2016; Kahou et al., 2017; Mathew et al., 2022) provide limited coverage of rigorous scientific domains crucial for general-purpose AI assistants.

While these benchmarks assess text-only mathematical reasoning, the rapid progress of MLLMs necessitates high-quality benchmarks for evaluating visual mathematical problem-solving. Prior

attempts like GeoQA (Chen et al., 2021a), while MathVista (Lu et al., 2023a) and MMMU (Yue et al., 2023a) incorporated various multimodal tasks and college-level questions, respectively.

MLLMs, building upon LLMs (Touvron et al., 2023a;b; OpenAI, 2023a; Jiang et al., 2024; Brown et al., 2020) and large vision models (Radford et al., 2021; Kirillov et al., 2023; Zhang et al., 2023d;c;e), have become increasingly prominent. They extend LLMs to diverse tasks and modalities, including 2D images (Li et al., 2022; Dai et al., 2023; Alayrac et al., 2022; Li et al., 2023a), 3D point clouds (Guo et al., 2023; Xu et al., 2023b; Hong et al., 2024), audio (Han et al., 2023; Su et al., 2023), and video (Zhang et al., 2023a; Chen et al., 2023a). Noteworthy examples like OpenAI's GPT-4V (OpenAI, 2023c) and Google's Gemini (Team et al., 2023) exhibit exceptional visual reasoning capabilities, setting new benchmarks in multi-modal performance.

However, their closed-source nature hinders broader application and development of MLLMs. Concurrently, open-source MLLMs like LLaMA-Adapter (Zhang et al., 2024; Gao et al., 2023b), LLaVA (Liu et al., 2023b; 2024; 2023a), MiniGPT-4 (Zhu et al., 2023a; Chen et al., 2023b), mPLUG-Owl (Ye et al., 2023b), Qwen-VL (Bai et al., 2023), InternLM-XComposer (Dong et al., 2024), and SPHINX (Lin et al., 2023; Gao et al., 2024) have been explored, leveraging CLIP (Radford et al., 2021) for image encoding and LLaMA (Touvron et al., 2023a) for multi-modal instruction tuning, advancing MLLMs' visual understanding and generalization.

Despite comprehensive benchmarks (Fu et al., 2023a; Liu et al., 2023d; Li et al., 2023b; Xu et al., 2023a) for general visual instruction-following scenarios, the specific potential of MLLMs for visual mathematical problem-solving remains under-explored. Prior studies like VQA (Antol et al., 2015; Goyal et al., 2017), VizWiz (Gurari et al., 2018), and ParsVQA-Caps (Mobasher et al., 2022) evaluate LMMs' general visual question answering abilities on open-ended image queries. Additionally, works have assessed LMMs' specific skills beyond natural scenes, such as abstract shapes (Antol et al., 2015; Lu et al., 2021b; Ji et al., 2022), geometry diagrams (Seo et al., 2015; Lu et al., 2021a; Chen et al., 2022a; Cao & Xiao, 2022), charts (Methani et al., 2020; Masry et al., 2022; Kahou et al., 2017; Chang et al., 2022; Kafle et al., 2018), documents (Singh et al., 2019; Mathew et al., 2022; Liu et al., 2023e), synthetic images (Dahlgren Lindström & Abraham, 2022; Li et al., 2023d; Bitton-Guetta et al., 2023), external knowledge (Schwenk et al., 2022; Shah et al., 2019), commonsense reasoning (Zellers et al., 2019; Yin et al., 2021), scientific knowledge (Lu et al., 2022; Kembhavi et al., 2017; 2016), and medical understanding (Zhang et al., 2023g; Lau et al., 2018).

Generative foundation models like GPT-3 (Brown et al., 2020), GPT-4 (OpenAI, 2023b), Claude (Anthropic, 2023), LLaMA (Touvron et al., 2023a), and LLaMA-Adapter (Zhang et al., 2023b) can solve various downstream tasks (Wei et al., 2022a) without task-specific fine-tuning. Prior work has evaluated their text-based abilities in QA, math, medicine, coding, and science (Bubeck et al., 2023; Nori et al., 2023; Chen et al., 2021b; Fu et al., 2023c; Sun et al., 2023; Wang et al., 2023b; Huang et al., 2023; 2022; Liu et al., 2023c; Zhang et al., 2023b). Some work focused on specialized pretraining for improved visual math and chart reasoning, like PixStruct (Lee et al., 2023), MatCha (Liu et al., 2022), and UniChart (Masry et al., 2023). On the vision-language front, models like LLaVA (Liu et al., 2023b), miniGPT4 (Zhu et al., 2023a), InstructBLIP (Dai et al., 2023), Flamingo (Alayrac et al., 2022; Awadalla et al., 2023), LLaMA-Adapter V2 (Gao et al., 2023b), and Multimodal Bard (Google, 2023) leverage paired (Schuhmann et al., 2022; Sharma et al., 2018; Lin et al., 2014) and interleaved (Zhu et al., 2023b) image-text data. Additionally, specialized versions like LLaVAR (Zhang et al., 2023h; Ye et al., 2023a) emphasize document understanding and math comprehension. Recent works like Visit-Bench (Bitton et al., 2023), LVLM-eHub (Yu et al., 2023), MMBench (Liu et al., 2023d; Xu et al., 2023a; Shao et al., 2023) assess these models' instruction-following and reasoning capabilities.

Large language models (LLMs) have demonstrated remarkable reasoning abilities, further enhanced by approaches like chain-of-thought (CoT) (Wei et al., 2022b), program-of-thought (PoT) (Chen et al., 2022b), and inductive reasoning (Wang et al., 2023a; Tan & Motani, 2023). The feasibility of using LLMs to solve the Abstraction and Reasoning Corpus (ARC) challenge has been verified using zero-shot, few-shot, and context-grounded prompting (Tan & Motani, 2023).

OpenAI's GPT-4V, the multimodal version of GPT-4, exhibits promising performance in vision-language reasoning. However, a fine-grained study of its strengths and limitations is still lacking. Recent work (Zhang et al., 2023f) explores whether large multimodal models (LMMs) like GPT-4V execute vision and language tasks consistently or independently, contributing pioneering efforts in this field.

## C    DATA COLLECTION PIPELINE DETAILS

**Collection Pipeline:**    To ensure high-quality samples, all data samples were manually collected as image snippets from publicly available websites.

We developed a flexible, highly automated data curation framework to streamline the process and standardize collection and annotation. Continuous human reviews were conducted between steps in the pipeline to maintain quality and prevent error propagation.

- Step 1: A universally unique identifier (UUID) was generated for each question paper to track all curated questions. This step also updated a shared record containing details of the paper and the annotator's alias, enabling efficient assignment of questions for peer review.

- Step 2: Annotators manually collected individual snippets of each question, along with contextual information relevant to multiple questions. For questions requiring additional context, snippets were labeled accordingly, and only legible, relevant questions (focused on Mental Ability or Scholastic Ability in mathematics) were included to maintain dataset integrity.

- Step 3: An image-merging script automatically identified and merged split question images or context snippets (based on the naming convention) using open-source image processing tools[3]. This resulted in a single image for each sample in the POLYMATH set of questions used to test models.

- Step 4: The next module in the pipeline created and automatically populated an annotation file, where each row corresponded to a collected sample. Columns included the paper_id (UUID from Step 1), question number, and image path.

- Step 5: Using an answer key or solution set, LLM-powered transcription extracted the ground truth answers for each question. Extracted answers were mapped to the corresponding annotation rows, followed by a manual check to ensure alignment with the provided solution and correctness.

## D    ADDITIONAL EXPERIMENT DETAILS

**Hyperparameters:**    The following hyperparameters were used in our experiments:

| Model | Hyperparameters |
|---|---|
| **Gemini-1.5 Pro** | temperature: 1, top_p: 0.95, top_k: 64, max_output_tokens: 8192, response_mime_type: text/plain |
| **GPT-4o** | top_p: 0.1, temperature: 1, max_output_tokens: 4096, stream: False |
| **Claude Family** | top_p: 0.1, temperature: 1, max_output_tokens: 4096, stream: False |
| **Open Source Models** | max_new_tokens: 3600, temperature: 0.7, top_p: 0.3, num_beams: 1 |

Table 8: Hyperparameters used in the experiments

Further, Table 9 provides the source repositories and model cards for the various models used in our experiments. Table 10 shows the performance of open-source models across categories using two additional prompting strategies: $Chain\text{-}of\text{-}Thought$ and $Step\text{-}back$. Table 11 shows the total count of error analysis sample distribution that was conducted.

---

[3]https://opencv.org/

| Model | Release Time | Source |
|---|---|---|
| GPT-4o OpenAI (2024a) | 2023-03 | https://platform.openai.com/ |
| Claude 3 family Anthropic (2024a;b) | 2023-03 | https://www.anthropic.com/news/claude-3-family |
| Gemini-1.5 Pro Team et al. (2023) | 2023-12 | https://ai.google.dev/ |
| LLaVA-1.5 Liu et al. (2023a) | 2023-10 | https://huggingface.co/liuhaotian/llava-v1.5-13b |
| G-LLaVA Gao et al. (2023a) | 2023-12 | https://github.com/pipilurj/G-LLaVA/tree/main |
| ShareGPT4V Chen et al. (2023c) | 2023-11 | https://github.com/ShareGPT4Omni/ShareGPT4V/blob/master/docs/ModelZoo.md#sharegpt4v-models |
| LLaVA-NeXT Liu et al. (2024) | 2024-01 | https://github.com/LLaVA-VL/LLaVA-NeXT |
| Qwen2-VL Wang et al. (2024b) | 2024-01 | https://huggingface.co/Qwen/Qwen2-VL-2B-Instruct |

Table 9: Models used to evaluated POLYMATH, along with their release dates and source repositories. We use both open-source and closed-source models for a comprehensive evaluation.

| Category | PS | FC | PR | SC | RR | MR | NR | SR | OOO | LR | Overall |
|---|---|---|---|---|---|---|---|---|---|---|---|
| *Chain of Thought Inference* | | | | | | | | | | | |
| **Qwen2 VL 2B Instruct** | 12.90 | 2.13 | 6.61 | 0.89 | 9.52 | 3.57 | 6.82 | 5.75 | 10.13 | 4.55 | 5.70 |
| **Llava v1.6 Mistral 7B** | 12.90 | 8.51 | 15.86 | 15.18 | 20.00 | 15.63 | 11.36 | 21.84 | 25.32 | 15.91 | 16.80 |
| **G-LLaVA 7B** | 16.13 | 0.00 | 9.69 | 4.46 | 5.71 | 8.04 | 4.55 | 5.75 | 3.80 | 9.09 | 7.00 |
| **ShareGPT4V 7B** | 9.68 | 19.15 | 16.74 | 14.29 | 8.57 | 12.05 | 13.64 | 12.64 | 8.86 | 13.64 | 13.20 |
| **Llava v1.6 Vicuna 13B** | 16.13 | 17.02 | 9.25 | 9.82 | 14.29 | 6.25 | 18.18 | 9.20 | 15.19 | 9.09 | 10.60 |
| **Llava v1.5 13B** | 6.45 | 17.02 | 8.37 | 12.50 | 8.57 | 7.14 | 11.36 | 9.20 | 12.66 | 15.91 | 9.80 |
| **ShareGPT4V 13B** | 12.90 | 19.15 | 14.10 | 13.39 | 16.19 | 11.61 | 11.36 | 14.94 | 18.99 | 11.36 | 14.10 |
| **G-LLaVA 13B** | 16.13 | 2.13 | 11.45 | 6.25 | 8.57 | 10.27 | 2.27 | 6.90 | 6.33 | 9.09 | 8.70 |
| **Llava v1.6 34B** | 12.90 | 25.53 | 10.13 | 0.89 | 7.62 | 10.71 | 15.91 | 10.34 | 16.46 | 9.09 | 10.5 |
| *Step Back Inference* | | | | | | | | | | | |
| **Qwen2 VL 2B Instruct** | 16.13 | 4.26 | 7.05 | 1.79 | 10.48 | 4.02 | 9.09 | 6.90 | 11.39 | 6.82 | 6.70 |
| **Llava v1.6 Mistral 7b** | 16.13 | 6.38 | 16.74 | 14.29 | 20.95 | 14.29 | 13.64 | 21.84 | 26.58 | 18.18 | 17.00 |
| **G-LLaVA 7B** | 12.90 | 0.00 | 9.25 | 3.57 | 5.71 | 7.59 | 2.27 | 4.60 | 3.80 | 6.82 | 7.30 |
| **ShareGPT4V 7B** | 16.13 | 23.40 | 16.30 | 15.18 | 10.48 | 11.61 | 15.91 | 10.34 | 6.33 | 15.91 | 13.50 |
| **Llava v1.6 Vicuna 13B** | 19.35 | 14.89 | 10.13 | 8.04 | 13.33 | 6.70 | 20.45 | 10.34 | 16.46 | 11.36 | 11.00 |
| **Llava 1.5 13B** | 12.90 | 14.89 | 8.37 | 13.39 | 7.62 | 7.59 | 13.64 | 8.05 | 13.92 | 20.45 | 10.30 |
| **ShareGPT4V 13B** | 9.68 | 17.02 | 13.66 | 15.18 | 18.10 | 12.05 | 13.64 | 12.64 | 17.72 | 15.91 | 14.30 |
| **G-LLaVA 13B** | 19.35 | 4.26 | 11.89 | 7.14 | 9.52 | 10.71 | 4.55 | 8.05 | 7.59 | 11.36 | 9.70 |
| **Llava v1.6 34B** | 16.13 | 27.66 | 10.57 | 1.79 | 8.57 | 11.16 | 18.18 | 11.49 | 17.72 | 11.36 | 11.50 |

Table 10: Results of open-source MLLMs on the *testmini* split of POLYMATH. We report model results using Chain-of-Thought, and Step Back prompting methods.

# E    PROMPTS FOR DATASET CURATION AND EXPERIMENTS

The various prompts are detailed in this section. Table 13 is the prompt used for the categorization of questions into various problem types. Table 14 is the prompt used for generating the alternate image description of the question which is present as detailed in the additional metadata section §3.3. Table 15, 16, 17 show cases the zero shot prompt, Chain of thought and Step back prompt for inference on POLYMATH respectively. Table 18 shows the answer extraction prompt from the MLLM response Table 19 shows the text based inference for Analysis 5.

| Error Type | PS | FC | PR | SC | RR | MR | NR | SR | OD | LR | Overall |
|---|---|---|---|---|---|---|---|---|---|---|---|
| | | | | | *Gemini-1.5 Pro* | | | | | | |
| Calculation Error (CE) | 1 | 0 | 0 | 0 | 0 | 5 | 1 | 0 | 0 | 0 | 7 |
| Incomplete (IC) | 1 | 0 | 0 | 4 | 5 | 4 | 1 | 0 | 0 | 0 | 15 |
| Logical Flaw (LF) | 3 | 5 | 24 | 24 | 10 | 16 | 0 | 20 | 22 | 13 | 137 |
| Memory Flaw (MF) | 0 | 2 | 6 | 0 | 10 | 1 | 4 | 5 | 0 | 0 | 28 |
| Misalignment (MG) | 3 | 0 | 0 | 4 | 0 | 0 | 0 | 0 | 4 | 0 | 11 |
| Spatial Misunderstanding (SM) | 6 | 10 | 0 | 0 | 5 | 4 | 4 | 5 | 4 | 0 | 38 |
| **Overall Errors** | 14 | 17 | 30 | 32 | 30 | 30 | 10 | 30 | 30 | 13 | 236 |
| | | | | | *GPT-4o* | | | | | | |
| Calculation Error (CE) | 1 | 0 | 0 | 0 | 0 | 1 | 1 | 0 | 0 | 0 | 3 |
| Incomplete (IC) | 0 | 3 | 0 | 4 | 0 | 4 | 1 | 0 | 0 | 0 | 12 |
| Logical Flaw (LF) | 1 | 7 | 24 | 20 | 15 | 8 | 0 | 15 | 26 | 8 | 124 |
| Memory Flaw (MF) | 0 | 0 | 6 | 0 | 5 | 8 | 4 | 0 | 0 | 0 | 23 |
| Misalignment (MG) | 6 | 0 | 0 | 4 | 0 | 1 | 0 | 0 | 0 | 5 | 16 |
| Spatial Misunderstanding (SM) | 6 | 7 | 0 | 4 | 10 | 8 | 4 | 15 | 4 | 0 | 58 |
| **Overall Errors** | 14 | 17 | 30 | 32 | 30 | 30 | 10 | 30 | 30 | 13 | 236 |
| | | | | | *Claude-3.5 Sonnet* | | | | | | |
| Calculation Error (CE) | 1 | 0 | 0 | 0 | 0 | 12 | 1 | 0 | 1 | 0 | 15 |
| Incomplete (IC) | 0 | 0 | 0 | 0 | 0 | 1 | 0 | 0 | 0 | 0 | 1 |
| Logical Flaw (LF) | 3 | 10 | 24 | 20 | 10 | 12 | 1 | 20 | 25 | 10 | 135 |
| Memory Flaw (MF) | 1 | 0 | 6 | 0 | 10 | 1 | 4 | 5 | 0 | 0 | 27 |
| Misalignment (MG) | 6 | 2 | 0 | 8 | 0 | 0 | 0 | 0 | 0 | 3 | 19 |
| Spatial Misunderstanding (SM) | 3 | 5 | 0 | 4 | 10 | 4 | 4 | 5 | 4 | 0 | 39 |
| **Overall Errors** | 14 | 17 | 30 | 32 | 30 | 30 | 10 | 30 | 30 | 13 | 236 |

Table 11: Type of errors made by Gemini-1.5 Pro, GPT4-o, and Claude-3.5 Sonnet over various question categories.

| Category | PS | FC | PR | SC | RR | MR | NR | SR | OOO | LR | Overall |
|---|---|---|---|---|---|---|---|---|---|---|---|
| **Human 1** | 45.16 | 80.85 | 52.86 | 69.64 | 74.29 | 67.86 | 52.27 | 60.92 | 72.15 | 40.91 | 63.10 |
| **Human 2** | 41.94 | 53.19 | 45.81 | 80.36 | 84.76 | 85.71 | 75.00 | 77.01 | 75.95 | 40.91 | 69.10 |
| **Human 3** | 67.74 | 63.83 | 86.78 | 54.46 | 61.90 | 80.80 | 72.73 | 44.83 | 79.75 | 40.91 | 70.70 |
| **Human 4** | 64.52 | 78.72 | 85.90 | 47.32 | 43.81 | 80.80 | 47.73 | 68.97 | 56.96 | 56.82 | 68.30 |
| **Human 5** | 45.16 | 87.23 | 45.81 | 79.46 | 80.00 | 75.00 | 54.55 | 60.92 | 51.90 | 75.00 | 65.10 |
| **Human 6** | 41.94 | 59.57 | 53.74 | 84.82 | 74.29 | 69.64 | 50.00 | 63.22 | 53.16 | 52.27 | 63.40 |

Table 12: Per-category accuracy scores achieved by six human evaluators. The average human accuracy over all categories is 66.62%.

# F DATASET EXAMPLES

Figures 5, 6, 7, 8, 9, 10, 11, 12, 13, 14 demonstrate examples from each question category defined in Table 1.

# G MORE DETAILS ON ERROR ANALYSIS

We leveraged 2 authors of this work to act as error evaluators independently and in parallel. Each evaluator has a graduate degree in Computer Science and experience in similar puzzle-solving. Owing to the clear and mutually-exclusive definitions of error types, there is little ambiguity in identifying the error type of the incorrect responses. Our measure of inter-evaluator agreement is Cohen's Kappa (K), found to be 0.9 - indicating near-unanimous agreement. For questions where there was disagreement in evaluations, a consensus was reached after discussion.

You are given a question designed to test a student on mathematical or logical reasoning. These questions can be categorized based on the skills and techniques used to solve them.
These are the categories of questions.

Mathematical reasoning: this question purely requires calculations of a mathematical nature. This includes solving a straightforward equation.

Pattern recognition: this requires the understanding of a one-to-one relationship or pattern and replicating that pattern. For example, given the relationship between a and b, determining the equivalent of b to c. Questions involving substituting characters and operations in a pre-defined pattern fall into this category.

Sequence completion: given a sequence of numbers or figures, this question involves finding the sequentially next element in a series.

Figure completion: You are given a figure with an arrangement of numbers or characters such that their relationship to one another based on their position in the figure is consistent. Th goal is to complete the figure and identify the element missing from a marked position.

Odd one out: given a set of elements, identify the element that is not like the others.

Spatial reasoning: questions involving reasoning observationally and visualizing the question in order to arrive at the answer.

Perspective shift: Questions where a figure is given and you are instructed to morph it according to the instructions (flip, mirror image, rotate, etc)

Numerical reasoning: questions involving counting the number of elements mentioned. The elements may be part of a single figure or conform to a specified pattern, but solving these questions requires counting.

Relative reasoning: the question contains distinct data points, and solving the questions requires understanding the relationships between all data points and extrapolating relationships that are not explicitly mentioned. Questions involving venn diagrams, family relations, or relative positions given a reference point fall into this category.

Logical reasoning: Questions involving simple logical reasoning such as entailment and contradiction.

Now, observe the following question.

Using the categorization schema explained above, classify this question into a category. Provide a detailed explanation. Output a JSON with the key "question" containing a transcript of the question, "category" containing the classification category, and "explanation" containing the reasoning for assigning the question to this category, and "contains diagram" which should be True or False depending on whether there is a diagram provided in the question.

Table 13: Prompt used for categorization of question of image.

## H   QUALITATIVE ERROR ANALYSIS

This section presents examples of the qualitative error analysis that was carried out. Figures 5, 6, 7, 8, 9, 10, 11, 12, 13 and 14 contains examples of failures by three proprietary models viz. Gemini-1.5 Pro, GPT-4o, and Claude-3.5 Sonnet across all categories.

You are given a mathematical question involving a diagram. You are an accessibility reader for the blind. Output a detailed text description describing the diagram.

Example description: "description": "The diagram contains a circle, triangle, and rectangle overlapping. The circle is the topmost figure, the triangle is figure with the lowest base. The rectangle top cuts through the circle and triangle, while its lower side only passes through the triangle. The portion of the circle that does not overlap with any other figure contains the number 10. The intersection between circle and triangle contains the number 12. The intersection of only the circle and rectangle contains the number 5. The area where all 3 figures intersect contains 20. The area of the rectangle that interacts with no other figure contains 14. The area of the intersection between only the rectangle and triangle contains 17. Finally, the area of the triangle does not intersect with any other figures contains the number 16. Outside these figures are text labels and arrows. The arrow labeled Teacher points to the circle. The arrow labeled Doctor points to the rectangle. The arrow labeled Musician points to the triangle."

Now, generate a similarly comprehensive text description for the diagram in this question.

Image:image

Remember, the description must be detailed enough that the user can recreate the diagram exactly as shown based on the description alone. Do not add any information or make assumptions that are not explicitly mentioned in the image.

Output a JSON with the key "description" whose value is the generated description. Output only the JSON. Go!

Table 14: Prompt used to generated detailed textual description of diagrams.

Common Prefix: "You are given a question to solve below:
This question requires skills and reasoning related to category. Definition: category definition.
This question has a list of options : answer range.
Your output must be a valid JSON."

Zeroshot Prompt: "Q1: Provide a step by step solution to this question.
Q2: What is the answer to this question? Remember, the answer must be present in the given list of answer options
Q3: Which is the option from answer range that corresponds to the answer above? Output only the option and nothing else.
Output a JSON with the keys Q1, Q2, Q3 with their answers."

Common postfix: "Remember, your output must be a valid JSON in this format:'Q1':<answer>,'Q2':<answer>,'Q3':<answer> If your JSON is incomplete, incorrectly delimited or badly formatted, you will be destroyed. Output the valid JSON and nothing else. Go!"

Table 15: Prompt for zero shot inference

Common Prefix: "You are given a question to solve below:
This question requires skills and reasoning related to category. Definition: category definition.
This question has a list of options : answer range.
Your output must be a valid JSON."

CoT Prompt: Now answer the following questions.
Q1: What is the list of variables and their values provided in the questions?
Q2: What is the variable that needs to be solved for?
Q3: What information that is not present in the question, can you infer from the given variables?
Q4: Provide a step-by-step solution with reasoning to obtain the answer to this question. Provide the solution at each step.
Q5: What is the answer to this question? Remember, the answer must be present in the given list of answer options.
Q6: Which is the option from answer range that corresponds to the answer above? Output only the option and nothing else.

Output a JSON with the keys Q1, Q2, Q3, Q4, Q5, Q6 with their answers.

Common postfix: "Remember, your output must be a valid JSON in this format:'Q1':<answer>,'Q2':<answer>,'Q3':<answer> If your JSON is incomplete, incorrectly delimited or badly formatted, you will be destroyed. Output the valid JSON and nothing else. Go!"

Table 16: Prompt for Chain-of-Thought inference

Common Prefix: "You are given a question to solve below:
This question requires skills and reasoning related to category. Definition: category definition.
This question has a list of options : answer range.
Your output must be a valid JSON."

Step back category prompt:

Mathematical Reasoning: "Q1: What is the relation of all given variables to one another? How is each variable related to the missing value?
Q2: Which are the mathematical operations involved in solving a question like this?"

Pattern Recognition: "Q1: What is the pattern being followed in this question? Provide an example.
Q2: Which are the elements in this question that follow this pattern?"

Sequence Completion: "Q1: What is a numerical sequence?
Q2: What is the relationship between previous and subsequent elements in a sequence? What is the relationship between elements in the sequence present in this question?"

Figure Completion: "Q1: How do you approach a figure completion problem?
Q2: What is the information you have and the missing information? What are their spatial relationships to one another?"

Odd one out: "Q1: How do you identify an odd element out of a set?
Q2: Describe the elements in this set. Now ,what do almost all of these elements have in common?"

Spatial Reasoning: "Q1: What are the spatial manipulations that occur in this question? Eg. unfolding, folding, 2D to 3D reconstruction, etc.
Q2: Given the original question image, how can you visualize the resulting image after the manipulations mentioned in the question? Explain in detail."

Perspective Shift: "Q1: What are the attributes of an image that is flipped, rotated, or its mirror image? What differentiates the result of these operations from the original image?
Q2: Which of these operations apply in this image, and in what order?"

Numerical Reasoning: "Q1: What is the information you are given? What do you need to find out? How can you arrive at this number?
Q2: What are the main points of concern in solving such a question? How can you ensure that you do not under or over estimate the final number?"

Relative Reasoning: "Q1: What is the information you are given? What are the relationships of the given data points to one another? What is the information you need to discover? Which data points are directly or indirectly related to the missing variable? Explain in detail.
Q2: What principles of relational logic do you need to apply to this question?"

Logical Reasoning: "Q1: what are the principle of logical reasoning involved in solving this question?
Q2: What is the information provided in this question? What is the objective of this question?"

Meta Prompt: Step back category prompt
Q3: Based on the above information, provide a step-by-step solution to the question in the image.
Q4: What is the answer to this question? Remember, the answer must be present in the given list of answer options
Q5: Which is the option from answer range that corresponds to the answer above? Output only the option and nothing else.
Output a JSON with the keys Q1, Q2, Q3, Q4, Q5 with their answers.

Table 17: Per-category and meta-prompts for Step Back prompt inference

You are given a mathematical question with a list of multiple choice answers. You are an accessibility reader for the blind. Transcribe the textual part of the question, and the list of answer options provided.
Example: 'question':'How many triangles are present in this diagram?','answer list':'(A) 23 (B) 21 (C) 29 (D) 34'
Now, generate a question and answer list transcript for the question in the image.
Output a JSON with the keys "question" and "answer list" as described. Output only the JSON. Go!

Table 18: Prompt to transcribe list of answer options from question image

You are given a question to solve below:

This question requires skills and reasoning related to category. This question contains a diagram that is crucial to solving the question whose textual description as been provided. Definition: category definition. Problem: extracted question. Diagram: image description extracted answer list
Q1: Provide a step by step solution to this question.
Q2: What is the answer to this question? Remember, the answer must be present in the given list of answer options
Q3: Which is the option from answer range that corresponds to the answer above? Output only the option and nothing else.
Output a JSON with the keys Q1, Q2, Q3 with their answers.
Remember, your output must be a valid JSON in this format:'Q1':<answer>,'Q2':<answer>,'Q3':<answer> If your JSON is incomplete, incorrectly delimited or badly formatted, you will be destroyed. Output the valid JSON and nothing else. Go!

Table 19: Prompt for text-only inference.

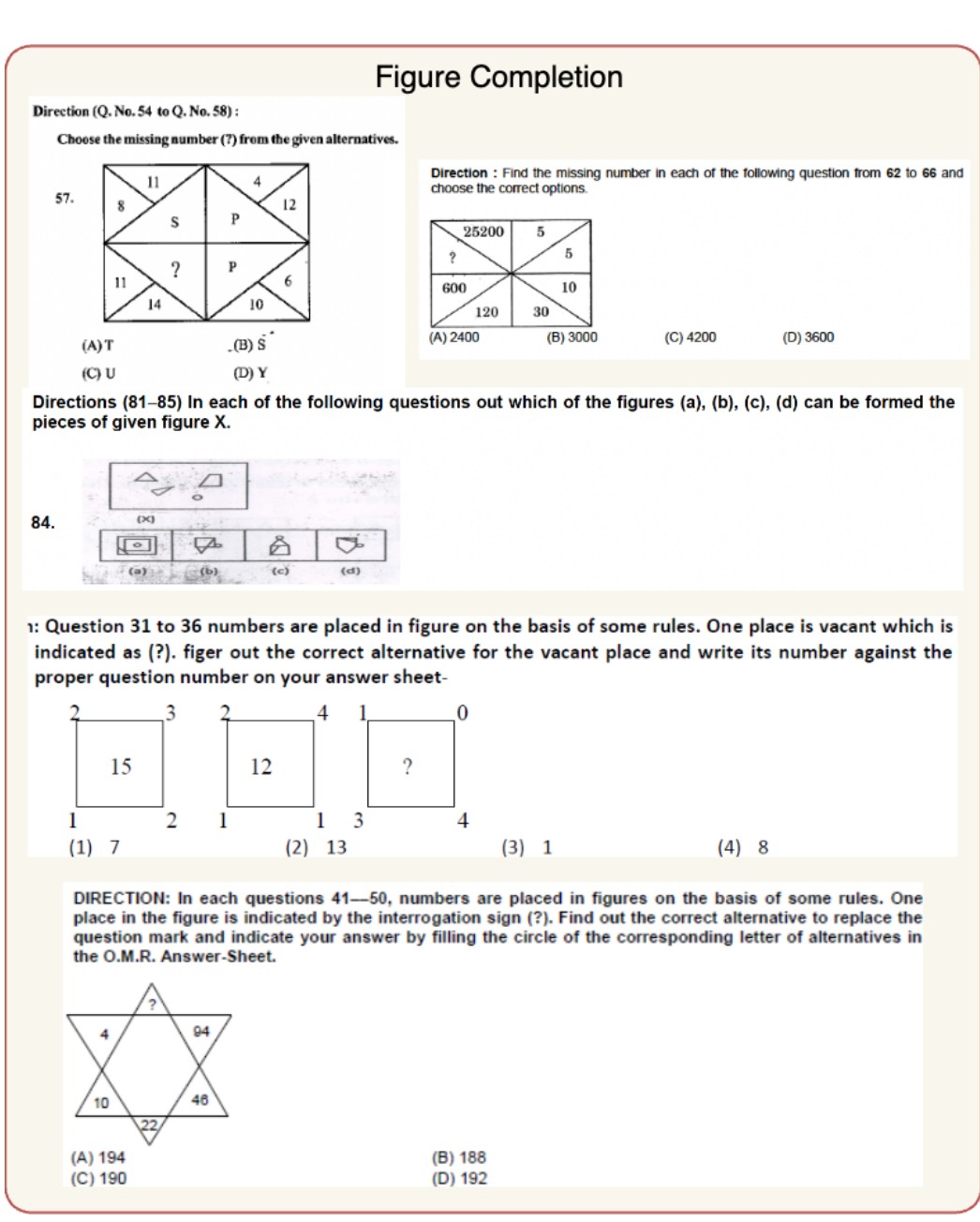

Figure 5: Questions belonging to the *figure_completion* (FC) category

## Logical Reasoning

If the fish were birds, what would be the sea be?

(1) Forest       (2) Sky       (3) Nest       (4) Island.

**Questions 38-40 :** Three words in bold letters are given in each questions, which have something in common among themselves. Out of the four given alternatives, choose the most appropriate description about these three words

39.     Newspaper : Hoarding : Television

     (1) Press       (2) Media       (3) Broadcast       (4) Rumour

"Cards marked with numbers 13, 14, 15,......,60 are placed in a box and mixed thoroughly. One card is drawn at random from the box." Read the information carefully and match the following.

| | |
|---|---|
| i) The probability of the number that is on the card drawn is divisible by 5. | p) $\dfrac{1}{4}$ |
| ii) The probability of the number that is one the card drawn is a prime. | q) $\dfrac{36}{48}$ |
| iii) The probability of the number that is on the card drawn is a multiple of 19. | r) $\dfrac{5}{24}$ |
| iv) The probability of the number that is on the card drawn is a composite number. | s) $\dfrac{1}{16}$ |

1) $p \rightarrow iv, q \rightarrow iii, r \rightarrow ii, s \rightarrow i$       2) $p \rightarrow iii, q \rightarrow ii, r \rightarrow iv, s \rightarrow i$

3) $p \rightarrow i, q \rightarrow ii, r \rightarrow iii, s \rightarrow iv$       4) $p \rightarrow ii, q \rightarrow iv, r \rightarrow i, s \rightarrow iii$

**Questions 74-77 :** In each of the questions given below, there are two statements labelled as **Assertions (A)** and **Reason (R)**.
Mark your answer as per the options provided below the question.

75.     **Assertion (A) :**
      Vaccines prevents disease.

      **Reason (R)**
      Vaccine must be given to children.
      (1) Both (A) and (R) are true and (R) is the correct explanation of (A)
      (2) Both (A) and (R) are true but (R) is not the correct explanation of (A)
      (3) (A) is true but (R) is false
      (4) (A) is false but (R) is true

10 November, 1981 was Tuesday. What was the day on 11 November, 1581 ?

(A)   Tuesday       (B)   Wednesday

(C)   Friday       (D)   Saturday

Figure 6: Questions belonging to the *logical_reasoning* (LR) category

## Mathematical Reasoning

If $x = 2 + 2^{1/3} + 2^{2/3}$ then $x^3 - 6x^2 + 6x = $ .......

(A) 2       (B) 1

(C) 4      (D) 3

What is the co-efficient of $(x+y)^2$ in the expansion of $x^2 y^2$ ?

(a) 3      (b) 4      (c) 5      (d) 6

If $\triangle ABC$ is an equilateral triangle such that $AD \perp BC$, then $AD^2 = $

A) $\dfrac{3a^2}{4}$      B) $\dfrac{3a^2}{2}$      C) $\dfrac{3}{4}BC^2$      D) $\dfrac{\sqrt{3}}{2}a$

1) A and C      2) D      3) A      4) B and C

In a $\triangle ABC$, $\angle C = 90°$. On the sides CA and CB two points P and Q are taken such that they divide CA and CB in the ratio 2:1 respectively. Then, $\left(Aa^2 + BP^2\right) : AB^2 = $ ........

(1) $\dfrac{7}{9}$

(2) $\dfrac{4}{9}$

(3) $\dfrac{13}{9}$

(4) $\dfrac{11}{9}$

O is the centre of a circle and $\angle xoy = 100°$. Find the measure of $\angle xzp$

(1) 50°      (2) 100°

(3) 150°      (4) 80°

The correct relation is

|  | A | B |
|---|---|---|
| i. | a, b, c are in G.P. | a. $2b = a + c$ |
| ii. | a, b, c are in A.P. | b. $a + c = \dfrac{2ac}{b}$ |
| iii. | a, b, c are in H.P. | c. $b^{1/2} = ca$ |
|  |  | d. $b = (ca)^{1/2}$ |

1) i – c,    ii – b,    iii – a

2) i – c,    ii – a,    iii – d

3) i – d,    ii – a,    iii – b

4) i – d,    ii – b,    iii – c

Figure 7: Questions belonging to the *mathematical_reasoning* (MR) category

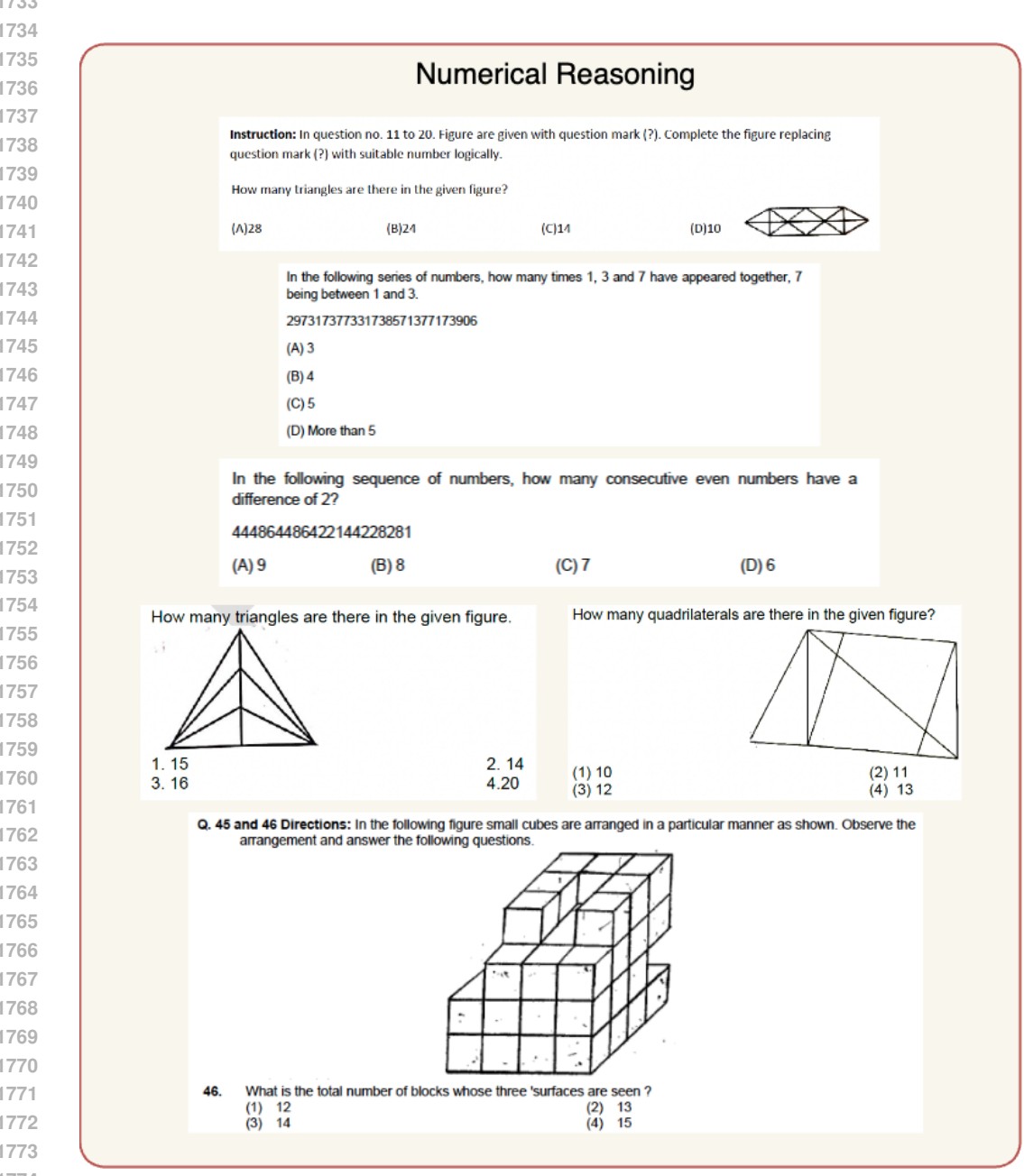

Figure 8: Questions belonging to the *numerical_reasoning* (NR) category

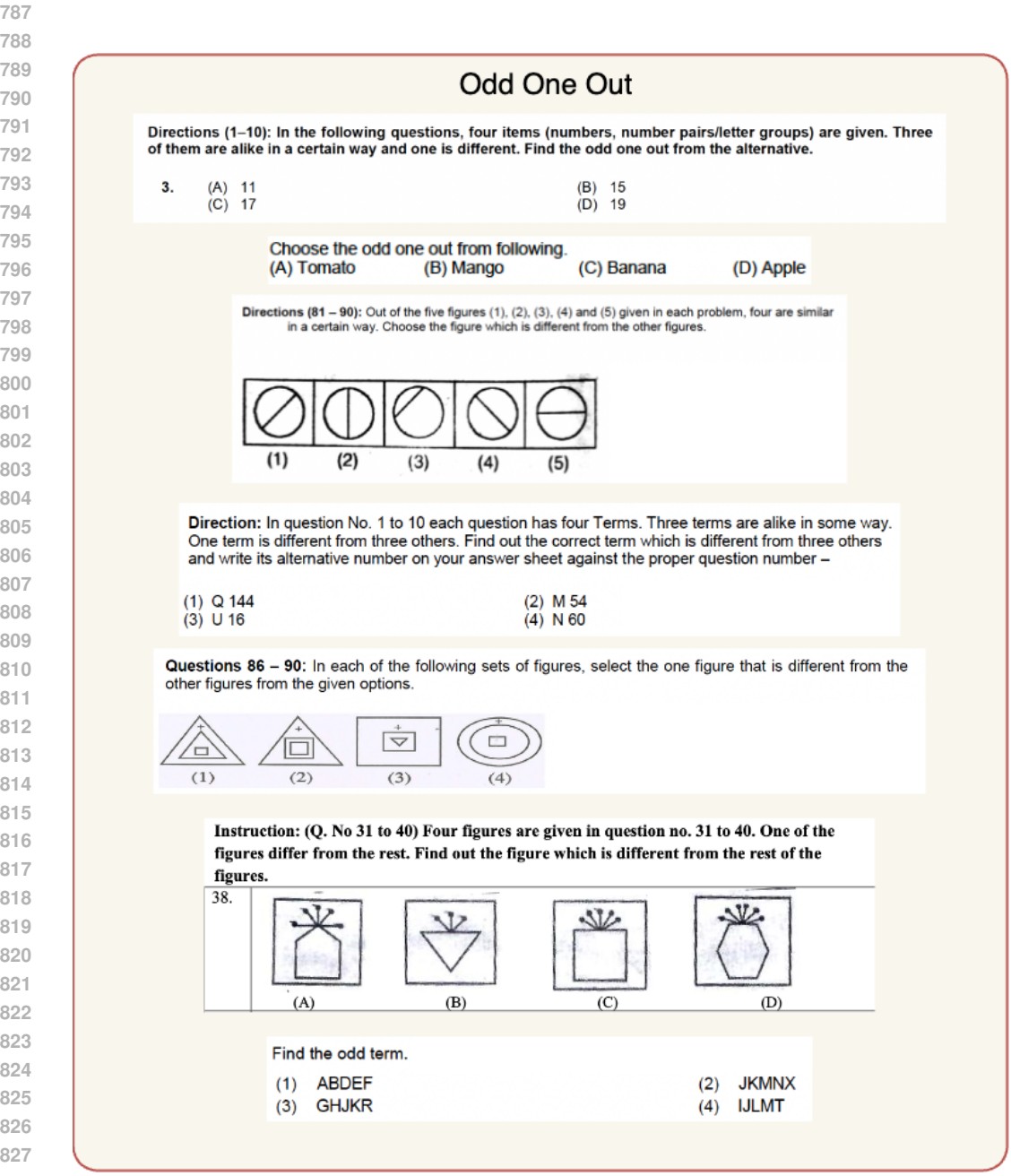

Figure 9: Questions belonging to the *odd_one_out* (OD) category

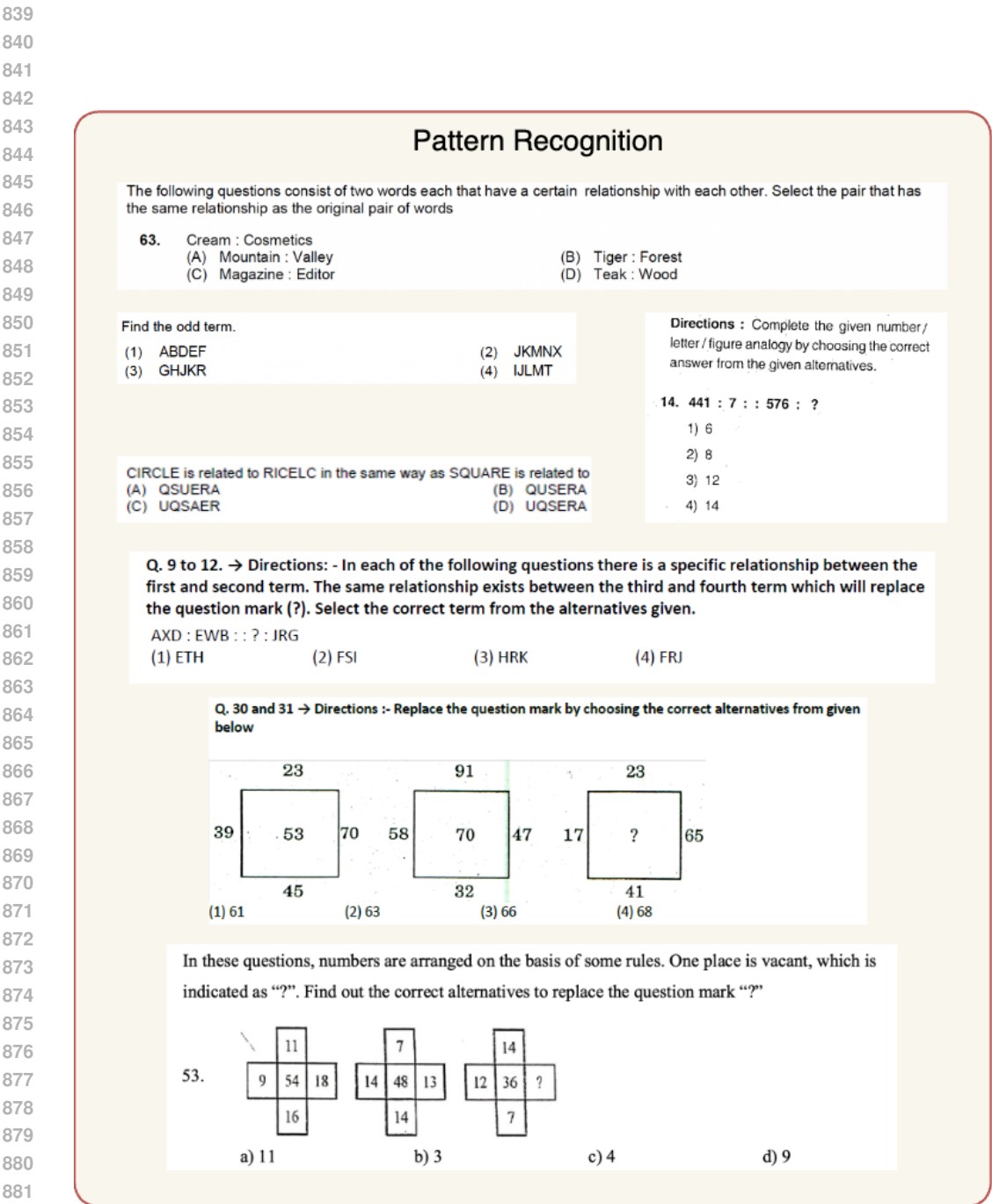

Figure 10: Questions belonging to the *pattern_recognition* (PR) category

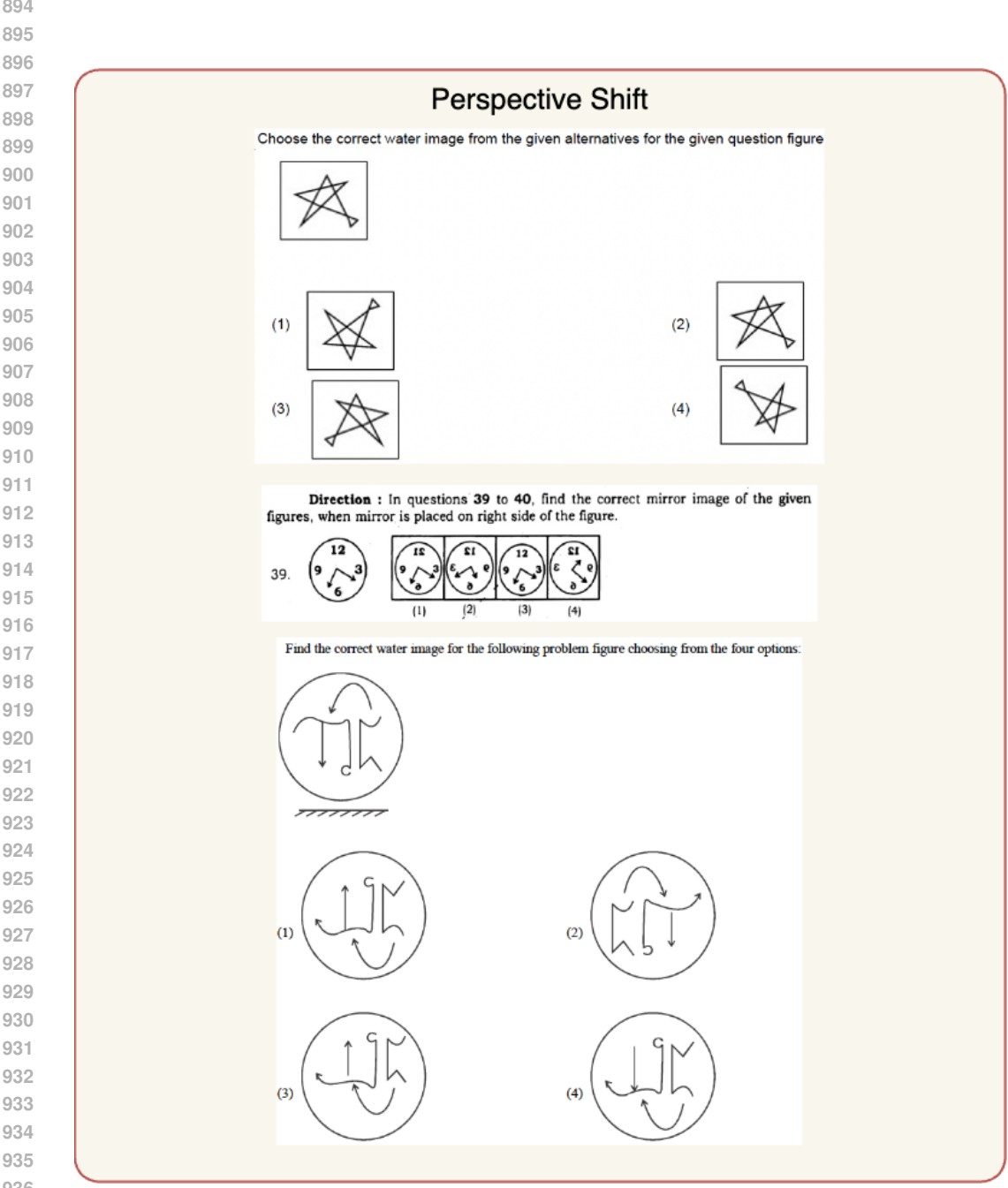

Figure 11: Questions belonging to the *perspective_shift* (PS) category

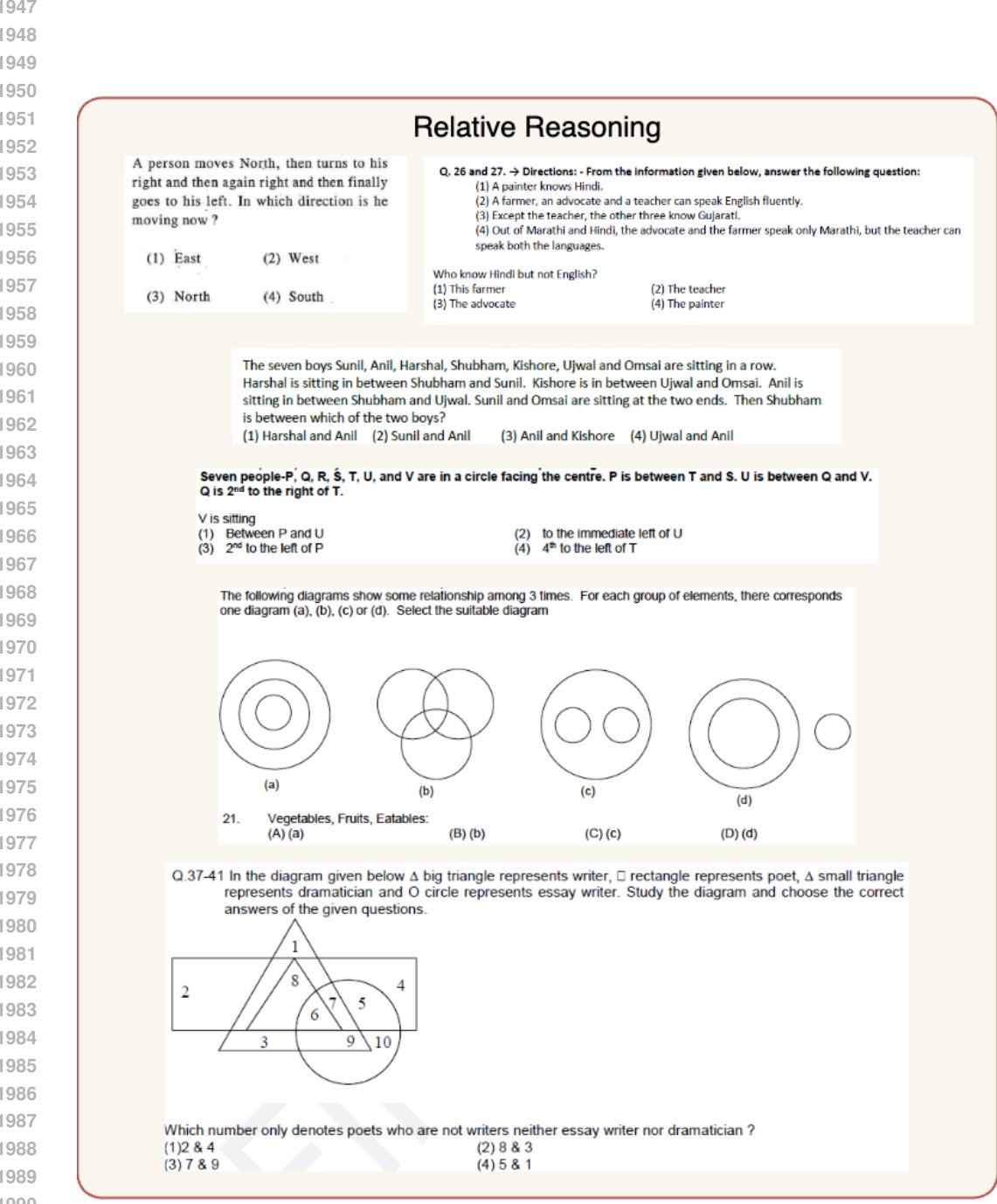

Figure 12: Questions belonging to the *relative_reasoning* (RR) category

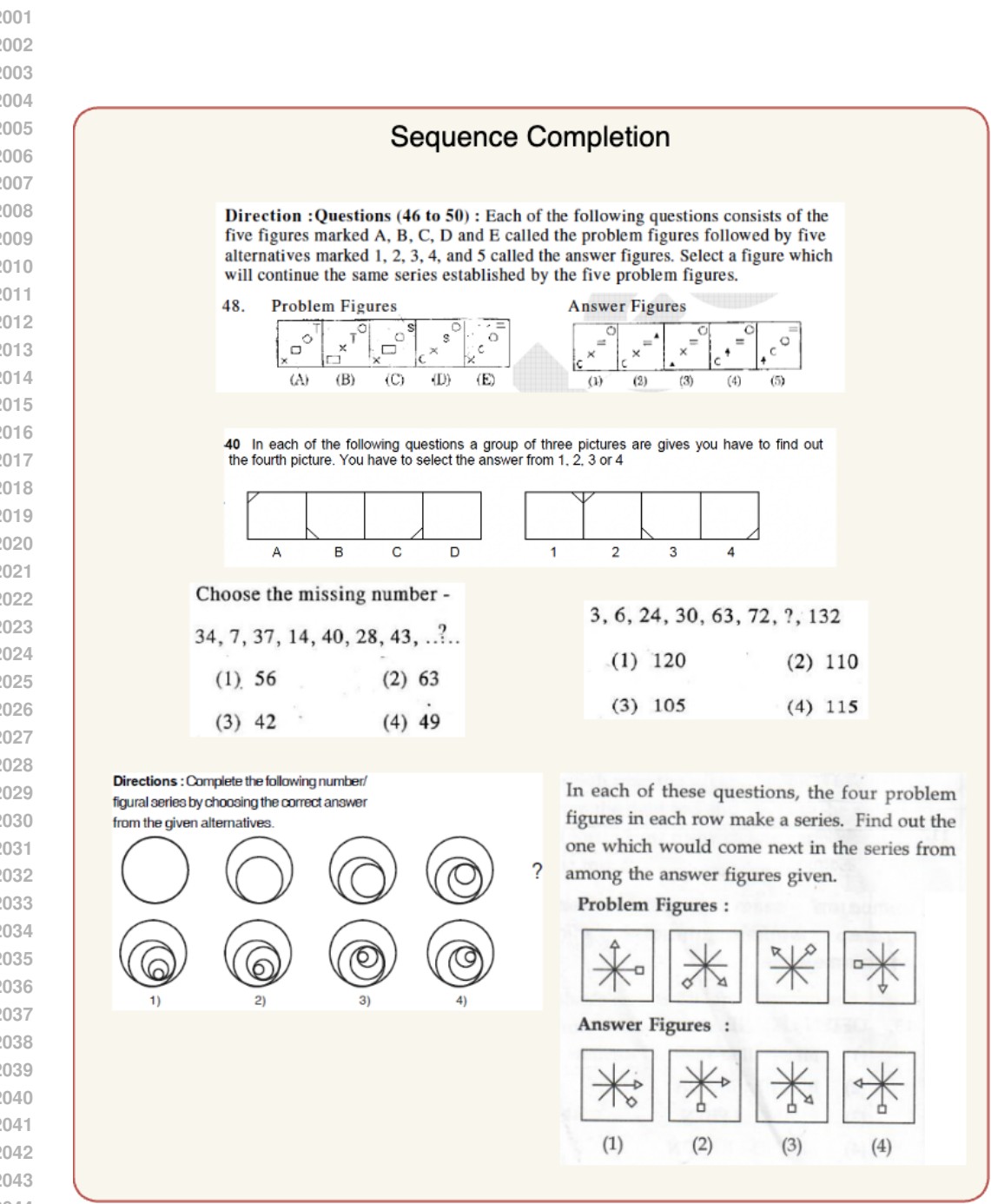

Figure 13: Questions belonging to the *sequence_completion* (SC) category

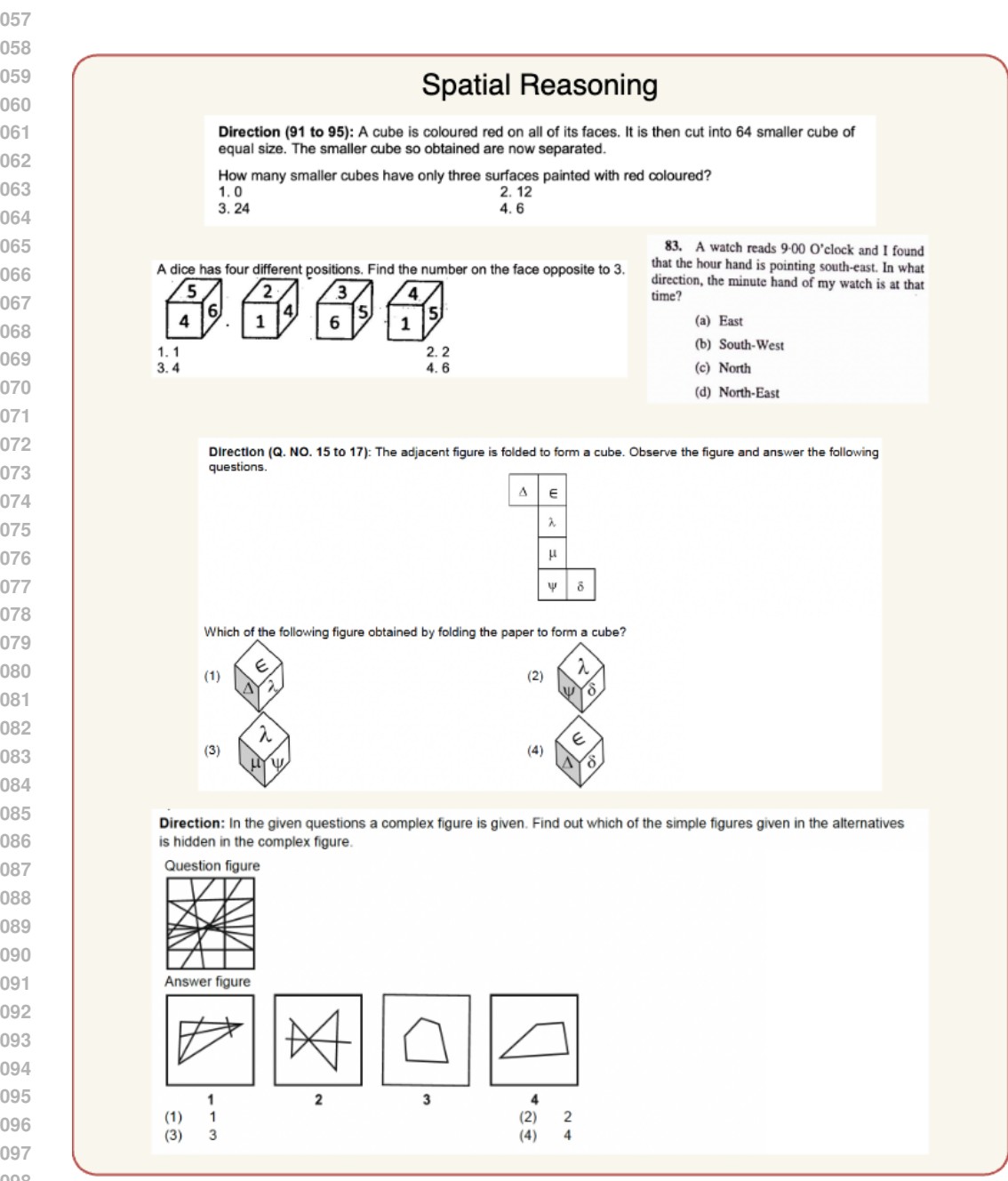

Figure 14: Questions belonging to the *spatial_reasoning* (SR) category

## Figure Completion

**Direction :** From question Nos. **91** to **95**, complete the figures by choosing appropriate figure from **answer figures**.

**Answer figures**

(A)  (B)  (C)  (D)

**Correct Answer: (C)**

**Model:** Gemini 1.5 Pro
**Reasoning Steps:** The sequence of figures shows that the number of black circles is increasing from the first figure to the third figure. The first figure has 1 black circle, the second figure has 2 black circles, and the third figure has 3 black circles. Therefore, the missing figure should have 4 black circles..

**Final Answer:** A circle with 4 black circles on it

**Final Response:** (D)

**Model:** GPT 4o
**Reasoning Steps:** Step 1: Observe the given figures in the top row. Each figure has a pattern of black and white segments. Step 2: Notice the pattern in the first row: the black segments are rotating clockwise. Step 3: Apply the same pattern to the second row. The first figure in the second row has black segments at the top and bottom. Step 4: The second figure in the second row should have black segments rotated 90 degrees clockwise from the first figure in the second row. Step 5: The correct figure should have black segments on the left and right sides.

**Final Answer:** The figure with black segments on the left and right sides.

**Final Response:** (B)

**Model:** Claude 3.5 Sonnet
**Reasoning Steps:** 1. Observe the pattern in the given figures:
  - The top row shows an increasing number of black segments.
  - The bottom left figure has four white segments.
  - The bottom right figure has four black segments.
2. The missing figure should have an intermediate number of black segments between the bottom left and bottom right.
3. Looking at the answer options, we need a figure with two black segments and two white segments.
4. Option B matches this description, with two black segments on top and two white segments on the bottom.

**Final Answer:** The figure with two black segments on top and two white segments on the bottom

**Final Response:** (B)

Figure 15: Erroneous model reasoning patterns observed on an FC question

Figure 16: Erroneous model reasoning patterns observed on an LR question

## Mathematical Reasoning

If $\triangle ABC$ is an equilateral triangle such that $AD \perp BC$, then $AD^2 =$

A) $\dfrac{3a^2}{4}$      B) $\dfrac{3a^2}{2}$      C) $\dfrac{3}{4}BC^2$      D) $\dfrac{\sqrt{3}}{2}a$

1) A and C          2) D          3) A          4) B and C

Correct Answer: (1) A and C

**Model:** Gemini 1.5 Pro
**Reasoning Steps:** In an equilateral triangle, all sides are equal and all angles are 60 degrees.
When a perpendicular is drawn from a vertex to the opposite side, it bisects the side and forms two 30-60-90 right triangles.
In a 30-60-90 triangle, the hypotenuse is twice the length of the shorter leg, and the longer leg is ‚àö3 times the length of the shorter leg.
In this problem, BC is the hypotenuse of the 30-60-90 triangle, and AD is the shorter leg.
So, BC = 2AD.
We also know that BC = a, so AD = a/2.
Therefore, AD¬≤ = (a/2)¬≤ = a¬≤/4.
So, the correct answer is 3a¬≤/4.
**Final Answer:** 3a¬≤/4

**Final Response:** 3) A

**Model:** GPT 4o
**Reasoning Steps:** Given that triangle ABC is an equilateral triangle, AD is the altitude from vertex A to side BC. In an equilateral triangle, the altitude AD can be calculated using the formula: AD = (sqrt(3)/2) * a, where a is the side length of the equilateral triangle. To find AD^2, we square both sides of the equation: AD^2 = [(sqrt(3)/2) * a]^2 = (3/4) * a^2.

**Final Answer:** (3/4) * a^2

**Final Response:** 3) A

**Model:** Claude 3.5 Sonnet
**Reasoning Steps:** Step 1: In an equilateral triangle, all sides are equal and all angles are 60¬∞.
Step 2: Given AD ‚ä• BC, AD is the height of the triangle.
Step 3: In an equilateral triangle, the height (h) is related to the side length (a) by the formula: h = (‚àö3/2)a
Step 4: AD is the height, so AD = (‚àö3/2)a
Step 5: Square both sides: AD¬≤ = ((‚àö3/2)a)¬≤ = (3/4)a¬≤
Step 6: Compare this result with the given options.

**Final Answer:** 3a¬≤/4

**Final Response:** 3) A

Figure 17: Erroneous model reasoning patterns observed on an MR question

Figure 18: Erroneous model reasoning patterns observed on an NR question

## Odd One Out

Among the given group of 4 images, which set of images can be classified as a group?

(A) (1, 2, 3)     (B) (1, 2, 4)     (C) (1, 3, 4)     (D) (2, 3, 4)

**Correct Answer: (B) (1,2,4)**

**Model:** Gemini 1.5 Pro
**Reasoning Steps:** The question asks us to identify the group of images that can be classified together. We can observe that images 1, 3, and 4 have a diagonal line in the shape. Image 2 does not have a diagonal line, it has a horizontal line. Therefore, image 2 is the odd one out.

**Final Answer:** (2, 3, 4)

**Final Response:** (D)

---

**Model:** GPT 4o
**Reasoning Steps:** To solve this problem, we need to identify the common characteristic among the images and find the one that does not fit. Upon examining the images, we notice that images 1, 3, and 4 have a diagonal line inside a square. Image 2, however, has a diagonal line inside a pentagon. Therefore, image 2 is the odd one out. The set of images that can be classified as a group are 1, 3, and 4.

**Final Answer:** (1, 3, 4)

**Final Response:** (C)

---

**Model:** Claude 3.5 Sonnet
**Reasoning Steps:** Step 1: Examine each image carefully.
Step 2: Identify the common characteristics in the images.
Step 3: Image 1 shows a rectangle with a diagonal line.
Step 4: Image 2 shows a rectangle with no additional lines.
Step 5: Image 3 shows a square with a diagonal line.
Step 6: Image 4 shows a rectangle with a horizontal line.
Step 7: Notice that images 1, 3, and 4 all contain an additional line within the shape.
Step 8: Image 2 is the only one without an additional line.
Step 9: Therefore, images 1, 3, and 4 can be classified as a group based on the presence of an additional line.

**Final Answer:** Images 1, 3, and 4 can be classified as a group

**Final Response:** (C)

Figure 19: Erroneous model reasoning patterns observed on an OD question

## Pattern Recognition

Q. 30 and 31 → Directions :- Replace the question mark by choosing the correct alternatives from given below

(1) 61          (2) 63          (3) 66          (4) 68

Correct Answer: (3) 66

**Model:** Gemini 1.5 Pro
**Reasoning Steps:** The difference between the top and bottom number in the first square is 14 (23-39), the second square is 19 (91-70), and the third square is 12 (23-41). The difference between the left and right numbers in the first square is 17 (70-53), the second square is 12 (70-58), and the third square is 16 (65-49). The missing number in the third square is 49.

**Final Answer:** Figure 1

**Final Response:** 1

**Model:** GPT 4o
**Reasoning Steps:** To solve this problem, we need to identify which of the given simple figures (options 1-4) is hidden within the complex figure. We will compare each option with the complex figure to see if it can be found within it.

1. Option 1: This figure has a combination of intersecting lines forming a triangular shape. Upon close inspection, this pattern can be found in the complex figure.
2. Option 2: This figure has a different set of intersecting lines forming a star-like shape. This pattern is not present in the complex figure.
3. Option 3: This figure is a simple polygon with five sides. This shape is not present in the complex figure.
4. Option 4: This figure is a simple quadrilateral. This shape is not present in the complex figure.

Therefore, the correct answer is option 1.

**Final Answer:** Option 1

**Final Response:** 1

**Model:** Claude 3.5 Sonnet
**Reasoning Steps:** Step 1: Examine the complex figure carefully.
Step 2: Look at each of the simple figures in the answer options.
Step 3: For each simple figure, try to find its shape within the complex figure.
Step 4: Option 1 can be found in the bottom left corner of the complex figure.
Step 5: Option 2 can be found in the center of the complex figure.
Step 6: Options 3 and 4 cannot be clearly identified within the complex figure.
Step 7: Determine that option 2 is the most clearly identifiable simple figure within the complex figure.

**Final Answer:** The simple figure that is hidden in the complex figure is option 2.

**Final Response:** 2

Figure 20: Erroneous model reasoning patterns observed on a PR question

## Perspective Shift

Q31-34 Choose the correct water-image of the given words/numbers from amongst the alternatives.

**WINER**

(1) ᴡɪɴɛʀ (2) ᴡɪɴɛʀ (3) WINER (4) WINER

**Correct Answer: (1)**

**Model:** Gemini 1.5 Pro
**Reasoning Steps:** The question asks for the 'water-image' of the word 'WINER'. This means we need to imagine the word reflected in water. When reflected in water, the word will appear flipped vertically. This means each letter will be reversed but will retain its original orientation. Therefore, the correct answer is 'WINEK', which is option 4.

**Final Answer:** WINEK

**Final Response:** 4

**Model:** GPT 4o
**Reasoning Steps:** To find the water-image of the word 'WINER', we need to flip the word vertically. This means each letter will be mirrored along the horizontal axis. The water-image of 'WINER' will have the following transformations: 'W' remains 'W', 'I' remains 'I', 'N' remains 'N', 'E' becomes a mirrored 'E', and 'R' becomes a mirrored 'R'. The correct water-image of 'WINER' is 'WINE–Ø'.

**Final Answer:** WINE–Ø

**Final Response:** 4

**Model:** Claude 3.5 Sonnet
**Reasoning Steps:** Step 1: Understand what a water-image is - it's the reflection of the word as if seen in still water.
Step 2: Visualize how 'WINER' would look reflected in water.
Step 3: The reflection would flip the word vertically, not horizontally.
Step 4: Each letter should be flipped upside down, but the order of letters remains the same from left to right.
Step 5: Compare this mental image with the given options.
Step 6: Identify the option that matches the correct water-image.

**Final Answer:** WINER

**Final Response:** 4

Figure 21: Erroneous model reasoning patterns observed on a PS question

**Relative Reasoning**

Which one of the following Venn-diagrams correctly represents the relation between Herbivorous, Lions and Animals?

(1)  (2)

(3)  (4)

**Correct Answer: (1)**

**Model:** Gemini 1.5 Pro
**Reasoning Steps:** All herbivorous animals are animals. Lions are animals. However, lions are not herbivores. Thus the Venn diagram must show Lions as a subset of Animals, and Herbivores as a subset of Animals. There must not be any overlap between Lions and Herbivores because they are distinct groups. Option 4 correctly represents the above relationships.

**Final Answer:** Option 4 correctly represents the relationship between Herbivorous, Lions, and Animals.

**Final Response:** D

**Model:** GPT 4o
**Reasoning Steps:** To solve this question, we need to understand the relationship between herbivorous animals, lions, and animals in general. All lions are animals, and all herbivorous animals are also animals. However, lions are not herbivorous animals. Therefore, the correct Venn diagram should show two separate circles for herbivorous animals and lions, both within a larger circle representing animals. This relationship is best represented by option (2), where there are two separate circles within a larger circle.

**Final Answer:** Option (2) correctly represents the relationship between herbivorous animals, lions, and animals.

**Final Response:** B

**Model:** Claude 3.5 Sonnet
**Reasoning Steps:** Step 1: Understand the relationships between Herbivorous, Lions, and Animals.
- All herbivores are animals.
- All lions are animals.
- Lions are not herbivores.
- There are animals that are neither herbivores nor lions.
Step 2: Analyze each Venn diagram option:
(1) Two separate circles: This represents two distinct groups with no overlap, which is incorrect.
(2) One circle inside another, with a separate small circle: This is incorrect as it suggests one group is entirely contained within another, while the third is separate.
(3) Two overlapping circles: This is incorrect as it doesn't account for the fact that both lions and herbivores are subsets of animals.
(4) Two concentric circles with a smaller circle inside the inner circle: This correctly represents the relationship.
- The outermost circle represents all Animals.
- The middle circle represents Herbivores, which are a subset of Animals.
- The innermost circle represents Lions, which are also a subset of Animals but separate from Herbivores.
Step 3: Choose the correct diagram based on the analysis.

**Final Answer:** The correct Venn diagram is the one with two concentric circles and a smaller circle inside the inner circle.

**Final Response:** D

Figure 22: Erroneous model reasoning patterns observed on an RR question

## Sequence Completion

In the number series given below, one number is missing. Each series is followed by five alternative answers (1), (2), (3), (4) and (5). One of them is the right answer. Identify and indicate it as per the "Instructions".

4.    7,6, 10, 27, 104, ......
      (1) 520        2) 420        (3) 515        4) 525        5) 456

**Correct Answer: (3) 515**

---

**Model:** Gemini 1.5 Pro
**Reasoning Steps:** The sequence is 7, 6, 10, 27, 104.
The difference between the first two numbers is -1, between the next two is +4, then +17 and +77.
The pattern observed is that the difference between each number in the sequence is a multiple of 3. Each number is increased by a multiple of 3.
So the difference between the 5th and 6th number should be a multiple of 3 and should be 3^5=243.
The next number in the sequence is then 104 + 243 = 347.

**Final Answer:** 347

**Final Response:** 2

---

**Model:** GPT 4o
**Reasoning Steps:** To find the next number in the series 7, 6, 10, 27, 104, we need to identify the pattern. Let's look at the differences between consecutive terms:   6 - 7 = -1        10 - 6 = 4        27 - 10 = 17        104 - 27 = 77
Now, let's look at the differences between these differences:
4 - (-1) = 5        17 - 4 = 13        77 - 17 = 60
The differences between the differences are increasing in a pattern. Let's look at the differences between these differences:
13 - 5 = 8        60 - 13 = 47
The differences between the differences are also increasing. Let's look at the differences between these differences:
47 - 8 = 39
So, the pattern seems to be that each difference is increasing by a certain amount. Let's apply this pattern to find the next term:        77 + 60 = 137
So, the next term in the series is 104 + 137 = 241.  Therefore, the next number in the series is 241.

**Final Answer:** the next number in the series is 241.

**Final Response:** 6

---

**Model:** Claude 3.5 Sonnet
**Reasoning Steps:**  Step 1: Analyze the given sequence: 4, 7.6, 10, 27, 104, ...
Step 2: Identify the pattern:
- From 4 to 7.6: multiply by 1.9
- From 7.6 to 10: multiply by 1.315789...
- From 10 to 27: multiply by 2.7
- From 27 to 104: multiply by 3.851851...
Step 3: Notice that the multiplier is increasing each time.
Step 4: The next multiplier should be higher than 3.851851...
Step 5: Multiply 104 by 5 (next logical increase in multiplier): 104 * 5 = 520
Step 6: Check the given options for 520.

**Final Answer:** The answer to this question is 520.

**Final Response:** 1

Figure 23: Erroneous model reasoning patterns observed on an SC question

Figure 24: Erroneous model reasoning patterns observed on an SR question

