# OpenReview forum: "PolyMATH: A Challenging Multi-Modal Mathematical Reasoning Benchmark"
_ICLR.cc/2025/Conference — Submitted to ICLR 2025_

### Official Review · Reviewer_8wsD · 2024-10-22

**Soundness:** 2
**Presentation:** 3
**Contribution:** 2
**Rating:** 5
**Confidence:** 3

**Summary:**

The paper present PolyMath, a benchmark evaluating visual abstraction reasoning consists of 5000 mannually collected images acrosss 10 categories. The author also extract testmini (1k) and test-image (with diagram) subset of the benchmark for diverse usage. A comprehensive experiment on testmini evaluating 15 MLLMs with four diverse prompting strategies. The overall results demonstrate a gap between current MLLMs and humans. A fine-grained error analysis reveals model's weak capability on spatial relaition and high-level reasoning. Further ablation study on providing aditional textual description shows that the main difficulty may be comprehensing visual diagrams.  In the end, the author also evaluated the newly introduced GPT-o1 and find that its performance align with human baseline. The author release their benchmark for public usage.

**Strengths:**

1. A extensive mannually created benchmark: PolyMATH consists 5k high quality images covering 10 patterns, allowing for a more thorough assessment of MLLMs' abilities across various abstract reasoning concepts. The benchmark can be further splited into mini subset for quick evaluation and diagram-only subset for visual-centric abstract evaluation.
2. A comprehensive evaluation: The author conduct a comprehensive evaluation over 15 MLLMs with four prompting approaches, demonstrating current MLLMs visual abstraction capabilities. Further ablation study provide insight on weakness of MLLMs (diagram understanding)
3. The paper already release the dataset for public use.

**Weaknesses:**

1. Benchmark Positioning:  The comparison between previous VQA or read-world dataset is not enough. I think there is a bunch of benchmarks related with visual abstraction reasoning (https://arxiv.org/abs/2202.10284). Is PolyMATH different than previous dataset or If it is just a combination of previous one. Can the evaluation on PolyMATH bring new insight compared to previous ones?
2. Pattern Design:  The paper mentions the cognitive reasoning and abstract reasoning. But it seems lack a sufficient elaboration on the why do we need to evaulate these patterns on MLLMs. It will be helpful to show practical application of these pattern in real world scenraio or relavent research from human coginition literature.
3. Value Justification: From the ablation study on diagram-only subset, the result showing the major issue blocking models abstract visual reasoning ability is the diagream comprehension. Can we understand the model have difficulties on percepting diagrams? If so, it may make the evaluation less insightful if the major bottleneck is in model's perception abilities instead of abstract reasoning ability. An experiment showing how much detail and vision information MLLMs can percept from diagream will be helpful.

**Questions:**

1. Based on the examples in Figures 1 and 3, it appears that the format of the questions and answer choices varies. For instance, in the first question of Figure 1, choices are labeled with letters (A, B, C, D) at the bottom, while in the second question, choices are labeled with numbers (1, 2, 3, 4) on the right side. I believe reorganizing all questions in a consistent format could improve the validity of the evaluation.
2. The dataset is unevenly distributed, with three out of ten patterns accounting for more than 40% of the baseline data, while some patterns represent only 3% of the data. Could the imbalanced data distribution lead to misleading conclusions in Section 4.2?
3. In Lines 462 to 466, simply comparing test-img and testmini makes the argument that "decreased accuracy on test-img is largely due to the presence of diagram-based problems" problematic. It's possible that test-img has a different distribution compared to testmini. For example, the questions in test-img may be more challenging, prompting designers to include diagrams for better understanding.
4. The authors adopt four prompting strategies for evaluating the MLLMs but only mention performance differences in one sentence (Lines 418-421). Could more insights or explanations be provided regarding these four strategies?
5. As the paper presents the complete PolyMATH dataset with 5,000 images, it seems unusual to report results on only 20% of them. Could the authors also provide results for the entire dataset? This would be beneficial for comparing new MLLMs more comprehensively in future research.


Overall, I appreicate the effort author and team paid on construct such a comprehensive datasets. I am willing to adjust my score if questions metioned previously can be solved and clarified.

---

> ### Author Response · Authors · 2024-11-24
> **Reviewer 8wsD - Response to Questions - 1/2**
>
> > Weakness 1: Benchmark Positioning: The comparison between previous VQA or read-world dataset is not enough. I think there is a bunch of benchmarks related with visual abstraction reasoning (https://arxiv.org/abs/2202.10284). Is PolyMATH different than previous dataset or If it is just a combination of previous one. Can the evaluation on PolyMATH bring new insight compared to previous ones?
>
>
> We thank the reviewer for bringing this paper to our notice. We would like to highlight that most of the benchmarks in the above mentioned survey are saturated. Plus these benchmarks focus on limited reasoning capabilities and contain significantly fewer samples. We would also like to draw attention towards some of the recent benchmarks that too [puzzleqa, marvel, mathverse, mathvista] covered just a few aspects of abstract reasoning capabilities but do not completely capture it. Polymath has been created from scratch and its details are mentioned in section 3.1. Given that most of the existing benchmarks are already saturated,
>
> > Weakness 2: Pattern Design: The paper mentions the cognitive reasoning and abstract reasoning. But it seems lack a sufficient elaboration on the why do we need to evaulate these patterns on MLLMs. It will be helpful to show practical application of these pattern in real world scenraio or relavent research from human coginition literature.
>
> Previous literature [1,2,3,4] has suggested cognitive and abstract reasoning are highly relevant for Artificial general intelligence. These skills are necessary for higher order thinking and are essential for basic reasoning abilities. They show that developed cognition and reasoning and understanding abilities are skills critical to success in a plethora of real world domains.
>
>
> > Weakness 3: Value Justification: From the ablation study on diagram-only subset, the result showing the major issue blocking models abstract visual reasoning ability is the diagream comprehension. Can we understand the model have difficulties on percepting diagrams? If so, it may make the evaluation less insightful if the major bottleneck is in model's perception abilities instead of abstract reasoning ability. An experiment showing how much detail and vision information MLLMs can percept from diagream will be helpful.
>
> While we do demonstrate that models, in opposition of their highly-vaunted multimodal capabilities, still have a long way to go in terms of visual comprehension - this weakness is by means the only, or even the major, cause of the failures observed on this dataset.
> We conduct a further, question category-wise study into the types of errors made by models. Specifically - we look into categories which have a high proportion of diagram based questions - where, if the lack of visual reasoning skills is truly a bottleneck, we would expect to see a high proportion of spatial misunderstanding errors.
> We present the results of the study below
>
> | Category          | % of questions with diagrams | % of questions with spatial misunderstanding errors |
> | ----------------- | ---------------------------- | --------------------------------------------------- |
> | Figure completion | 100                          | 28% (sonnet) - 55% (gemini)                         |
> | Perspective shift | 66                           | 25% (sonnet) - 50% (gemini, gpt4o)                  |
> | Spatial reasoning | 55                           | 16% (gemini, sonnet) - 50% (gpt4o)                  |
>
>
> This contradicts the hypothesis, forcing us to conclude that while visual reasoning is an area with a significant scope for improvement, this dataset poses challenges in logical and abstract reasoning and that range beyond testing only visual comprehension of an image.
>
> Citations:
> 1. Zhao, Jian, et al. "Cognitive psychology-based artificial intelligence review." Frontiers in Neuroscience 16 (2022): 1024316.
> 2. Lohman, David F., and Joni M. Lakin. "Intelligence and reasoning." The Cambridge handbook of intelligence (2011): 419-441.
> 3. Clement, Bradley J., Edmund H. Durfee, and Anthony C. Barrett. "Abstract reasoning for planning and coordination." Journal of Artificial Intelligence Research 28 (2007): 453-515.
> 4. Wang, Yingxu, et al. "Cognitive intelligence: Deep learning, thinking, and reasoning by brain-inspired systems." International Journal of Cognitive Informatics and Natural Intelligence (IJCINI) 10.4 (2016): 1-20.

---

> ### Author Response · Authors · 2024-11-24
> **Reviewer 8wsD - Response to Questions - 2/2**
>
> > Question 1: Based on the examples in Figures 1 and 3, it appears that the format of the questions and answer choices varies. For instance, in the first question of Figure 1, choices are labeled with letters (A, B, C, D) at the bottom, while in the second question, choices are labeled with numbers (1, 2, 3, 4) on the right side. I believe reorganizing all questions in a consistent format could improve the validity of the evaluation.
>
> Thank you for your feedback. The evaluation pipeline is doing it consistently but we will take this change this in the final version of the dataset.
>
> > Question 2: The dataset is unevenly distributed, with three out of ten patterns accounting for more than 40% of the baseline data, while some patterns represent only 3% of the data. Could the imbalanced data distribution lead to misleading conclusions in Section 4.2?
>
> We recognise the fact that the benchmark is imbalanced. We are working on increasing the benchmark as a continous effort comparable to prev benchmark datasets. We're actively involved in improving the breadth and depth of data.
>
> > Question 3: "decreased accuracy on test-img is largely due to the presence of diagram-based problems" problematic. It's possible that test-img has a different distribution compared to testmini. For example, the questions in test-img may be more challenging, prompting designers to include diagrams for better understanding.
>
> Thank you for the question. We want to highlight that some questions are purely diagrammatic in nature, without any question that requires a clarifying diagram like the Perspective Shift category (Fig. 11 page 36 and Fig. 21 page 46). As the difficulty across categories is fairly balanced as they are meant to equitably assess different cognitive skills, test-img includes figure-based questions spanning the range from simplistic to challenging - reflecting the difficulty of the testmini and the larger test dataset. We also illustrate that spatial misunderstanding is a significant driver of errors, which further supports our observation on the limited perceptual abilities of MLLMs.
>
> > Question 4: The authors adopt four prompting strategies for evaluating the MLLMs but only mention performance differences in one sentence (Lines 418-421). Could more insights or explanations be provided regarding these four strategies?
>
> Thank you for the suggestion. We found the following insights: Step-back and chain of thought give longer responses, are more cohesive, and have similar scores via different models. Step-back is more vulnerable to memory issues and misalignment, as the “stepping back” approach results in increased distance from the problem and possibly reduces focus on the particulars of the problem and answer options. For categories where the principle behind the question requires more thought than the specifics of the question (eg. perspective shift), step back emerges as the best strategy; for categories where the approach to the problem depends on the details of the questions themselves (eg. figure completion), CoT exhibits better overall performance. We also see model preference patterns emerge; for example, few-shot inference is particularly conducive to high performance when using Claude 3.5 Sonnet.
>
> > Question 5: As the paper presents the complete PolyMATH dataset with 5,000 images, it seems unusual to report results on only 20% of them. Could the authors also provide results for the entire dataset? This would be beneficial for comparing new MLLMs more comprehensively in future research
>
> Thank you for your feedback. Please find the results on the entire test set. We will include them in the final draft of the paper. We followed the convention of MathVista to give the results on a smaller test set in the main paper.
>
> | Row Labels        | FC    | LR    | MR    | NR    | OD    | PR    | PS    | RR    | SC    | SR    | Overall |
> | ----------------- | ----- | ----- | ----- | ----- | ----- | ----- | ----- | ----- | ----- | ----- | ------- |
> | Gemini 1.5 Pro    | 24.03 | 37.27 | 34.61 | 30.00 | 36.27 | 27.11 | 16.34 | 30.29 | 29.39 | 32.49 | 30.68   |
> | GPT 4o            | 22.32 | 55.91 | 34.61 | 40.91 | 47.86 | 27.82 | 19.61 | 32.00 | 25.81 | 47.37 | 34.16   |
> | Cllaude 3 sonnet  | 21.46 | 43.64 | 25.51 | 32.73 | 33.25 | 25.09 | 22.88 | 33.14 | 27.60 | 27.23 | 28.06   |
> | Claude Haiku      | 19.31 | 24.09 | 20.87 | 28.64 | 30.23 | 23.50 | 23.53 | 20.57 | 25.09 | 22.43 | 23.28   |
> | Claude 3.5 sonnet | 29.18 | 79.09 | 35.59 | 50.91 | 53.65 | 27.82 | 45.75 | 32.76 | 31.18 | 51.03 | 38.42   |

---

> > ### Author Response · Authors · 2024-11-27
> > **Follow up on the rebuttal**
> >
> > Dear Reviewer,
> >
> > We trust this message finds you well. We are writing to follow up on the rebuttal response we submitted in response to your feedback. We are eager to receive your insights and comments on our response. If there are any areas that you still find unclear or require further clarification, we would be more than happy to provide additional information to ensure a comprehensive understanding of our work.
> >
> > Your feedback is of utmost importance to us, and we greatly appreciate your time and consideration in evaluating our research.
> >
> > Thank you for your attention, and we look forward to hearing from you soon.

---

> > > ### Comment · Reviewer_8wsD · 2024-11-27
> > >
> > > I am satisfied with the author's response, and I also check other's reviewer's comments. I will keep my rating as I am still concerned about questions 1 and 2. Based on the author's rebuttal, these two questions seem not easily solvable, at least in the current submission (e.g. **The evaluation pipeline is doing it consistently, but we will take this change in the final version of the dataset.**  **We recognise the fact that the benchmark is imbalanced. We are working on increasing the benchmark as a continuous effort comparable to previous benchmark datasets.**).

---

> > > > ### Author Response · Authors · 2024-11-28
> > > > **Response to Reviewer 8wsD's comment**
> > > >
> > > > We thank the reviewer for their feedback.
> > > >
> > > > Q1 We would like to emphasise that our evaluation logic (inference_models/claude.py line 191 of the anonymous github link) normalises all categorical values between A to D and 1 to 4 to a common 1-4 scale, using a mapping function. This ensures that our evaluation pipeline uses consistent answer options, and is extendable to previously unseen configurations as well.
> > > >
> > > >
> > > > Q2. For the 2nd part of the response, we showcase the results of the entire test set by randomly selecting 153 samples (minimum number of samples for any of the 10 categories). The results on this sampled test set is very similar to the actual test set and test mini presented in the paper. We will include these results in the final version of the paper:
> > > >
> > > > |            | FC    | LR    | MR    | NR    | OD    | PR    | PS    | RR    | SC    | SR    | Overall |
> > > > | ---------- | ----- | ----- | ----- | ----- | ----- | ----- | ----- | ----- | ----- | ----- | ------- |
> > > > | gemini     | 27.45 | 33.99 | 35.95 | 28.76 | 32.68 | 23.53 | 13.07 | 33.33 | 29.39 | 33.99 | 29.21   |
> > > > | gpt 4o     | 19.61 | 58.82 | 32.68 | 43.14 | 45.75 | 30.07 | 22.88 | 28.76 | 24.18 | 48.37 | 35.42   |
> > > > | sonnet 3   | 19.61 | 45.75 | 22.22 | 35.95 | 35.95 | 22.22 | 19.61 | 36.60 | 26.14 | 28.76 | 29.28   |
> > > > | haiku      | 22.88 | 20.92 | 19.61 | 30.07 | 28.76 | 24.84 | 22.22 | 21.57 | 26.14 | 20.92 | 23.79   |
> > > > | 3.5 sonnet | 26.14 | 82.35 | 38.56 | 47.71 | 55.56 | 26.80 | 49.02 | 29.41 | 30.07 | 52.29 | 43.79   |
> > > >
> > > >
> > > > We would also like to draw the reviewers’ attention to some previous popular benchmarks [1-8] that were imbalanced but are very relevant for measuring LLMs performance. Since our benchmark was created from NTSE question papers, the category distribution reflects the distribution that was present in the papers itself. We would like to add that we are actively working on sourcing more data that would mitigate the imbalance, as part of our ongoing effort to upgrade this dataset.
> > > >
> > > >
> > > > 1. Goyal, Yash, et al. "Making the v in vqa matter: Elevating the role of image understanding in visual question answering." Proceedings of the IEEE conference on computer vision and pattern recognition. 2017.
> > > > 2. Singh, Amanpreet, et al. "Towards vqa models that can read." Proceedings of the IEEE/CVF conference on computer vision and pattern recognition. 2019.
> > > > 3. Mathew, Minesh, Dimosthenis Karatzas, and C. V. Jawahar. "Docvqa: A dataset for vqa on document images." Proceedings of the lIEEE/CVF winter conference on applications of computer vision. 2021.
> > > > 4. Mathew, Minesh, et al. "Infographicvqa." Proceedings of the IEEE/CVF Winter Conference on Applications of Computer Vision. 2022.
> > > > 5. Lei, Jie, et al. "Tvqa: Localized, compositional video question answering." arXiv preprint arXiv:1809.01696 (2018).
> > > > 6. Lu, Pan, et al. "Mathvista: Evaluating mathematical reasoning of foundation models in visual contexts." arXiv preprint arXiv:2310.02255 (2023).
> > > > 7. Rohrbach, Anna, et al. "Movie description." International Journal of Computer Vision 123 (2017): 94-120.
> > > > 8. Hiippala, Tuomo, et al. "AI2D-RST: A multimodal corpus of 1000 primary school science diagrams." Language Resources and Evaluation 55 (2021): 661-688

---

> > > > > ### Author Response · Authors · 2024-12-02
> > > > > **Follow up to the rebuttal**
> > > > >
> > > > > Dear Reviewer,
> > > > >
> > > > > We hope this email finds you well. We are following up on our rebuttal response to your valuable feedback on our research. As a reminder, the deadline for rebuttal responses is tomorrow, Dec 2nd. We would be grateful if you could share any further insights or comments you may have on our response at your earliest convenience. We truly appreciate your time and thoughtful review. Your feedback is crucial to improving our work, and we value your expertise in this field.

---

### Official Review · Reviewer_uu8E · 2024-11-01

**Soundness:** 2
**Presentation:** 2
**Contribution:** 2
**Rating:** 5
**Confidence:** 3

**Summary:**

This paper introduces POLYMATH, a multimodal mathematical reasoning benchmark designed to assess the performance of large multimodal language models (MLLMs) in terms of visual understanding and abstract reasoning abilities. The benchmark covers 10 different categories, including cognitive text and visual challenges such as pattern recognition, spatial reasoning, and relative reasoning. The researchers conducted a comprehensive and quantitative evaluation of 15 MLLMs using four different prompting strategies, including Chain-of-Thought and Step-Back.

**Strengths:**

1. A new multimodal mathematical reasoning benchmark, POLYMATH, is proposed, covering 10 different categories and contributing to the advancement of mathematics reasoning tasks.
2. The paper conducted a comprehensive and quantitative evaluation of 15 MLLMs using four different prompting strategies.

**Weaknesses:**

1. The paper does not provide open-source examples from the POLYMATH dataset. Without such examples, the unique challenges and value of the dataset are not clearly conveyed, weakening the paper’s effectiveness.
2. The analysis provided is superficial and covers mostly well-known conclusions without exploring the performance differences across models in different reasoning tasks or the underlying reasons. This shallow analysis fails to offer new insights or robust support for the conclusions, limiting the paper's contribution to the field.
3. Although the paper mentions the issue of large models’ mathematical reasoning capabilities, it lacks further exploration of this topic. The absence of an in-depth discussion or future directions weakens the paper's forward-looking impact and completeness.

**Questions:**

1. Till the deadline of the review process, the link to the benchmark is not open-source. I hope the authors would organize the benchmark website properly during the rebuttal so that the benchmark’s composition and evaluation content could be inspected.
2. Experiment analysis is a little superficial. I hope the authors may provide some detailed analysis of the experimental results to yield more insightful conclusions.
3. The paper proposes many problems of current LLM models, but does not provide possible solutions. It will make this paper more meaningful if the authors would do some explorations on it.

---

> ### Author Response · Authors · 2024-11-24
> **Reviewer uu8E - Response to Weakness - 1/2**
>
> >Weakness 1: The paper does not provide open-source examples from the POLYMATH dataset. Without such examples, the unique challenges and value of the dataset are not clearly conveyed, weakening the paper’s effectiveness.
>
> Please revisit the anonymous github link which contains details of the dataset along with examples: [Anonymous Github](https://anonymous.4open.science/r/PolyMATH-052D/README.md). Please download the repository as a zip since the website does not support png type format.
>
> > Weakness 2: The analysis provided is superficial and covers mostly well-known conclusions without exploring the performance differences across models in different reasoning tasks or the underlying reasons. This shallow analysis fails to offer new insights or robust support for the conclusions, limiting the paper's contribution to the field.
>
> Thank you for your feedback. We would like to reiterate the focus of our work which is to introduce a cognitive and spatial reasoning benchmark. We showcase that even current state of the art models suffer from problems which test their abstract and cognitive reasoning capabilities. While there have been similar studies ([1], [2], [3], [4]), our work is more holistic and covers a very broad range of problems. We have also conducted experiments with 15 state of the art models with 4 well known prompting strategies and are detailed in section 4.2 and 4.3. Although we get some similar findings as compared to some previous works, we showcase these findings on significantly different problem types. We are happy to conduct any further experiments or analysis as per reviewers’ suggestions.
>
> > Weakness 3: Although the paper mentions the issue of large models’ mathematical reasoning capabilities, it lacks further exploration of this topic. The absence of an in-depth discussion or future directions weakens the paper's forward-looking impact and completeness.
>
> In terms of future directions, we enumerate possible extensions to this work and the insights it contains in Section A of the Appendix. Primarily, the findings related to visual comprehension and comparative benchmarking invite further exploration into the differences between model capabilities, as well as concerted efforts to improve visual reasoning beyond the performance baselines we exhibit in this work. We believe that the various reasoning errors and categorical strengths and weaknesses examined in this work could provide a direction for future model development efforts.
> Additionally, our experiments involving generating textual descriptions and the observed improvement in performance, while exposing the shortcomings of multimodal reasoning, open up avenues for research into improving the performance of contemporary MLLMs by setting up robust inter-modal frameworks and creatively reframing problems in a way that best leverages these models’ strengths and mitigates their weaknesses.
> We believe these are productive/feasible lines of inquiry for future research building on this work.
>
> Citations:
>
> [1] Liu, Yuan, et al. "Mmbench: Is your multi-modal model an all-around player?." European Conference on Computer Vision. Springer, Cham, 2025
>
> [2] Li, Bohao, et al. "Seed-bench: Benchmarking multimodal llms with generative comprehension." arXiv preprint arXiv:2307.16125 (2023)
>
> [3] Fu, Chaoyou, et al. "A challenger to gpt-4v? early explorations of gemini in visual expertise." arXiv preprint arXiv:2312.12436 (2023)
>
> [4] Sun, Keqiang, et al. "Journeydb: A benchmark for generative image understanding." Advances in Neural Information Processing Systems 36 (2024)

---

> ### Author Response · Authors · 2024-11-24
> **Reviewer uu8E - Response to Questions - 2/2**
>
> > Q1: Till the deadline of the review process, the link to the benchmark is not open-source. I hope the authors would organize the benchmark website properly during the rebuttal so that the benchmark’s composition and evaluation content could be inspected.
>
> Please find the [Anonymous Github Repo](https://anonymous.4open.science/r/PolyMATH-052D/README.md)
>
> > Q2: Experiment analysis is a little superficial. I hope the authors may provide some detailed analysis of the experimental results to yield more insightful conclusions.
>
> Thanks for the feedback. We will add the additional analysis on O1, a large language model, demonstrates varying levels of performance across different cognitive tasks, ranging from those closest to human capabilities to those furthest away. In terms of pattern recognition and mathematical reasoning, O1-Mini closely approximates human performance. When it comes to relative and logical reasoning, O1-Preview is closer to human levels. Sequence completion and spatial reasoning pose the greatest challenges for O1, with both O1-Mini and O1-Preview falling significantly short of human performance. This disparity underscores a common limitation in less-performant language models and O1: difficulties with perception and visual comprehension. Interestingly, O1-Mini outperforms O1-Preview in numerical reasoning by approximately 20%, suggesting that scaling up the model size doesn't always translate to better performance in all tasks.
> We also did a thorough analysis of model errors and have added additional results for response R1. We are happy to conduct any further experiments or analysis as per reviewers’ suggestions.
>
> > Q3: The paper proposes many problems of current LLM models, but does not provide possible solutions. It will make this paper more meaningful if the authors would do some explorations on it.
>
> This dataset is intended as a benchmark to challenge researchers in the domain to improve future models by evaluating them on this dataset. We also identify errors that affect models so they can become a focus for improvement. To establish the efficacy of existing performance-improvement prompting methods, we report baseline results to provide a springboard for further advancements.

---

> ### Comment · Reviewer_uu8E · 2024-11-24
>
> I have reviewed the benchmark links you provided, and I find them acceptable. Regarding the questions about the LLMs, the authors have provided additional explanations, which are beneficial to the significance of this paper. These clarifications have addressed some of my concerns, so I will raise my score.
> As for comments on the novelty and depth of some conclusions in the paper, I have also referred to the feedback from other reviewers. I appreciate the authors' efforts in creating the dataset, and it would be even better if there were more impactful take-away findings.

---

> > ### Author Response · Authors · 2024-11-25
> > **Response to Reviewer uu8E's comment**
> >
> > Dear Reviewer,
> >
> > We trust this message finds you well. Thank you so much for increasing the score for our paper.  Please let us know if there are any additional experiments/analysis you would like us to conduct and we can get back to you.
> >
> > Your feedback is of utmost importance to us, and we greatly appreciate your time and consideration in evaluating our research.
> >
> > Thank you for your attention, and we look forward to hearing from you soon.

---

### Official Review · Reviewer_KHqw · 2024-11-02

**Soundness:** 3
**Presentation:** 3
**Contribution:** 2
**Rating:** 6
**Confidence:** 4

**Summary:**

This paper introduce PolyMath, a multimodal math benchmark curated from India National Talent Search Examination.
PloyMath is carefully curated following 5 steps on collection, categorization and quality assurance.
Extensive experiemnts are conducted, showing that the best MLLM only achieve at most 41.9% performance on PloyMath.
Further analysis find that MLLM rely more on textual descriptions than visual information, and the logical flaw to be the most common error.
Study also finds that openai-o1 model performs closely with human baseline on textual only version of the dataset.

**Strengths:**

- The writing is clear and easy to follow.
- The proposed dataset is challenging, which is useful for future development of LLMs and MLLMs.
- The ayalysis showing that MLLMs rely more on text descriptions and the detailed error analysis are insightful.

**Weaknesses:**

- The findings in the paper are not new. Firstly, for 'MLLMs rely more on text than image', MathVerse[1] specifically create 6 versions of their dataset with incremental reliance on visual information, showing the same trend.
- The error analysis suggest logical flaw to be the main bottle neck of MLLM's performance. But Table3 and 4 suggests that different inference stretagy does not help much, comparing with zero-shot. (only about 2%-6%) So it's unclear how we can improve our models on this dataset.
- Insufficient discussion on related works. Since PolyMath is scoped as abstract visual reasoning benchmark, the author should discuss how PloyMath is different from and better than existing benchmarks in this field, like Puzzle-VQA[2], Marvel[3].

[1] Zhang, R., Jiang, D., Zhang, Y., Lin, H., Guo, Z., Qiu, P., ... & Li, H. (2025). Mathverse: Does your multi-modal llm truly see the diagrams in visual math problems?. In European Conference on Computer Vision (pp. 169-186). Springer, Cham.

[2] Chia, Y. K., Han, V. T. Y., Ghosal, D., Bing, L., & Poria, S. (2024). PuzzleVQA: Diagnosing Multimodal Reasoning Challenges of Language Models with Abstract Visual Patterns. arXiv preprint arXiv:2403.13315.

[3] Jiang, Y., Zhang, J., Sun, K., Sourati, Z., Ahrabian, K., Ma, K., ... & Pujara, J. (2024). MARVEL: Multidimensional Abstraction and Reasoning through Visual Evaluation and Learning. arXiv preprint arXiv:2404.13591.

**Questions:**

- Table 5 and table 3's performance are not consistant, the reviewer understands they are on different subsets, but the big difference is not expected. Could the author explain why?
- For the second weakness above, could the author give hypothesis on why inference scaling doesn't help the model much? And what is expected to improve our model on PolyMath?
- Since the error analysis suggest logical flaw to be the main bottle neck of MLLM's performance (which is irrelavant to image information), and all data can be converted to textual format. What make it necessary for this dataset to be made multimodal?

---

> ### Author Response · Authors · 2024-11-24
> **Reviewer KHqw - Response to Weakness - 1/2**
>
> Thanks for the feedback and suggestions. We address each weakness and question below:
>
> > Weakness 1: The findings in the paper are not new. Firstly, for 'MLLMs rely more on text than image', MathVerse[1] specifically create 6 versions of their dataset with incremental reliance on visual information, showing the same trend.
>
> We would like to highlight that the core contribution of this paper is the challenging benchmark, which includes problems related to spatial and cognitive reasoning. These problems are essential for assessing the cognitive reasoning abilities of MLLMs, in contrast to those presented in MathVerse [1], which primarily focus on problems of plane geometry, solid geometry, and functions. Our main findings (i.e., lower performance on PolyMATH) indicate the limitations of MLLMs' ability to comprehend the reasoning aspects required to solve many complex real-world problems. For the other findings, we believe they align with those from MathVerse [1], suggesting that PolyMATH also assesses mathematical reasoning abilities beyond cognitive reasoning.
>
> > Weakness 2: The error analysis suggest logical flaw to be the main bottle neck of MLLM's performance. But Table3 and 4 suggests that different inference stretagy does not help much, comparing with zero-shot. (only about 2%-6%) So it's unclear how we can improve our models on this dataset.
>
> Thank you for this comment. We performed some preliminary experiments with various inference strategies, which resulted in marginal improvements. This suggests that off-the-shelf methods are not sufficient for achieving higher performance, highlighting the complexity of our proposed benchmark. However, we believe that our findings and insights from these experiments can help design better inference-time or alignment strategies to enhance the reasoning abilities of LLMs. Additionally, in alignment techniques involving preference optimization, preference data could be created to favor outputs with greater reasoning correctness. Furthermore, exploring newer techniques such as DPO and KTO for their potential to improve reasoning could be an interesting future direction.
>
> > Weakness 3: Insufficient discussion on related works. Since PolyMath is scoped as abstract visual reasoning benchmark, the author should discuss how PloyMath is different from and better than existing benchmarks in this field, like Puzzle-VQA[2], Marvel[3].
>
> We thank the reviewer for bringing the above mentioned papers to our attention. We will add them to our literature review section. Puzzle- VQA focuses on visual perception on MLLMs such as colors, numbers, sizes and shape. Marvel on the other hand focuses on Object Core Knowledge, Number Core Knowledge and Geometry Core Knowledge. While PuzzleQA and MARVEL study key aspects of abstract reasoning, our work focuses on cognitive reasoning capabilities on MLLMs which have not been studied by previous works in detail. Our benchmark covers broader categories like Perspective Shift, Figure Completion, Pattern Recognition, Sequence Completion, Relative Reasoning, Mathematical Reasoning, Numerical Reasoning, Spatial Reasoning, Odd One Out and Logical Reasoning. These categories of abstract and cognitive reasoning have not been studied in previous works. We would also like to highlight that our benchmark is significantly larger than the above mentioned benchmarks (2000, 770 samples vs. 5000 samples) and foundation models obtain nearly the same scores on all three benchmarks.
>
>
> [1] Zhang, R., Jiang, D., Zhang, Y., Lin, H., Guo, Z., Qiu, P., ... & Li, H. (2025). Mathverse: Does your multi-modal llm truly see the diagrams in visual math problems?. In European Conference on Computer Vision (pp. 169-186). Springer, Cham.
>
> [2] Chia, Y. K., Han, V. T. Y., Ghosal, D., Bing, L., & Poria, S. (2024). PuzzleVQA: Diagnosing Multimodal Reasoning Challenges of Language Models with Abstract Visual Patterns. arXiv preprint arXiv:2403.13315.
>
> [3] Jiang, Y., Zhang, J., Sun, K., Sourati, Z., Ahrabian, K., Ma, K., ... & Pujara, J. (2024). MARVEL: Multidimensional Abstraction and Reasoning through Visual Evaluation and Learning. arXiv preprint arXiv:2404.13591.

---

> ### Author Response · Authors · 2024-11-24
> **Reviewer KHqw - Response to Suggestions - 2/2**
>
> > Q1: Table 5 and table 3's performance are not consistant, the reviewer understands they are on different subsets, but the big difference is not expected. Could the author explain why?
>
> Test mini is a stratified sampling representation of the original benchmark which has roughly 600 questions which do not contain diagrams. On the other hand, test img has 1000 samples, all of which contain diagrams. The difference is due to the fact that MLLMs have greater difficulty in understanding questions with diagram.
>
> >Q2: For the second weakness above, could the author give hypothesis on why inference scaling doesn't help the model much? And what is expected to improve our model on PolyMath?
>
> We conducted an additional experiment where we provided diagram description, along with diagram image and question which resulted in a better performance. The results of the additional experiment are as follows:
>
> |                   | FC    | LR    | MR    | NR    | OD    | PR    | PS    | RR    | SC    | SR    | Overall |
> | ----------------- | ----- | ----- | ----- | ----- | ----- | ----- | ----- | ----- | ----- | ----- | ------- |
> | claude-3-haiku    | 23.40 | 25.00 | 24.55 | 18.18 | 35.44 | 29.52 | 48.39 | 28.57 | 40.18 | 34.48 | 30.00   |
> | gpt4o_accuracy    | 42.55 | 52.27 | 40.18 | 45.45 | 35.44 | 46.26 | 70.97 | 33.33 | 53.57 | 60.92 | 45.60   |
> | gemini_1_5_pro    | 44.68 | 54.55 | 41.96 | 59.09 | 56.96 | 28.63 | 29.03 | 42.86 | 38.39 | 37.93 | 40.50   |
> | claude-3-sonnet   | 42.55 | 47.73 | 36.16 | 50.00 | 62.03 | 29.52 | 29.03 | 42.86 | 38.39 | 37.93 | 39.00   |
> | claude-3-5-sonnet | 46.81 | 56.82 | 42.86 | 59.09 | 37.97 | 47.14 | 83.87 | 36.19 | 58.04 | 63.22 | 49.00   |
>
>
> > Q3: Since the error analysis suggest logical flaw to be the main bottle neck of MLLM's performance (which is irrelavant to image information), and all data can be converted to textual format. What make it necessary for this dataset to be made multimodal?
>
> The main aim is to study the ability of MLLMs’ ability to comprehend visual aspects to improve their mathematical reasoning, hence the multi-modal nature of this benchmark can be helpful in this aspect since this ability is not systematically studied yet. Based on our error analysis, we draw the following insights
> * Spatial misunderstanding is the second most common type of error
> * Logical errors are not limited to text only questions, and can also occur in image based questions due to surface level comprehension of the image and lack of abstract reasoning over the information provided.
> * Furthermore, the dataset includes some particularly complex images on which a conversion to text results in models reasoning incorrectly - hence converting the entire dataset to text cannot guarantee that the amount and integrity of information in image only questions will be preserved.
>
>
> In light of this, we aim for this benchmark to be used in a multimodal fashion. Furthermore:
> The diagrams’ text description was human verified and the subset on which this study was performed included image based questions whose diagrams could be converted into textual description without loss in information and comprehension. This was an ablation study, which showcased that even with textual descriptions the models perform far below human capabilities. The main intent of the benchmark is to evaluate multimodal reasoning capabilities itself.

---

> > ### Comment · Reviewer_KHqw · 2024-11-25
> >
> > Thanks for the detailed response, I'm generally satified with the response and I will increase my rating to 6.
> >
> > Please make sure you make the updates above, including the literature review, results when providing diagram description. Furthermore, it would be good to test different inference-time stretagy in such setting, so that we will have better idea how to improve the performance.

---

> > > ### Author Response · Authors · 2024-11-25
> > > **Response to Reviewer KHqw's comment**
> > >
> > > Dear Reviewer,
> > >
> > > We trust this message finds you well. Thank you so much for increasing the score for our paper. We have conducted a case study  with OpenAI O1 models for the same. Did your suggestion of "test different inference-time stretagy" meant testing more models like those ?  Please let us know if that is the proposed experiment/analysis you would like us to conduct and we can get back to you.
> > >
> > > Your feedback is of utmost importance to us, and we greatly appreciate your time and consideration in evaluating our research.

---

> > > > ### Comment · Reviewer_KHqw · 2024-11-25
> > > >
> > > > Thanks for the response, by inference-time stretagy I mean applying stretegies like CoT, ToT or PoT, under the diagram description experiment. (the table you presented above)
> > > >
> > > > Experiments with o1 models are good but since we can not give it multimodal yet and their thoughts are hidden, the inight from that is somewhat limited.

---

### Official Review · Reviewer_X95T · 2024-11-04

**Soundness:** 2
**Presentation:** 3
**Contribution:** 2
**Rating:** 6
**Confidence:** 3

**Summary:**

The paper introduces POLYMATH, a novel benchmark designed to evaluate the mathematical and visual reasoning capabilities of multi-modal large language models (MLLMs). The authors conducted extensive evaluations on 15 state-of-the-art MLLMs, analyzing their performance through multiple prompting strategies, such as zero-shot and Chain-of-Thought prompting. Results reveal significant gaps in MLLM performance compared to human baselines, particularly in spatial reasoning and diagram interpretation.

**Strengths:**

S1: The authors provide a detailed categorization schema, covering ten reasoning types, which ensures a thorough assessment of MLLM capabilities in cognitive tasks.

S2: The authors implemented a rigorous evaluation using multiple inference strategies, including zero-shot, few-shot, Chain-of-Thought, and Step Back prompting.

S3: A fine-grained analysis highlights common pitfalls in model reasoning, such as logical flaws and spatial misunderstandings, providing insights into areas that need improvement.

**Weaknesses:**

W1: While the paper presents error analysis for major models like Claude-3.5 Sonnet and GPT-4o, extending this analysis to more open-source models could yield deeper insights into shared weaknesses.

W2: The reported improvements when replacing diagrams with text descriptions raise questions about MLLM comprehension of visual information. A more in-depth discussion or qualitative examples would enhance understanding.

**Questions:**

The paper notes a performance improvement when diagrams were replaced by textual descriptions, indicating that MLLMs may not be effectively processing visual information. Could the authors provide more qualitative examples or specific insights into how the textual descriptions differed from the diagrams and why these led to improved performance?

---

> ### Author Response · Authors · 2024-11-24
> **Reviewer X95T - Response to Weakness & Suggestions**
>
> Thanks for the feedback and suggestions. We address each weakness and question below:
>
> > Weakness 1: While the paper presents error analysis for major models like Claude-3.5 Sonnet and GPT-4o, extending this analysis to more open-source models could yield deeper insights into shared weaknesses.
>
> Qwen2 VL (2B) Instruct (5) - Least performant open source model
>
> | Row Labels      | Percent       |
> |-----------------|---------------|
> | calc error      | 3.81  |
> | incomplete      | 0             |
> | logical flaw       | 63.55  |
> | memory          | 2.11  |
> | misalign        | 0.84  |
> | spatial         | 29.66   |
>
> LLaVA-v1.6 Mistral (7B) (15) - Best performing open source model
> | Row Labels      | Percent       |
> |-----------------|---------------|
> | calc error      | 2.54  |
> | incomplete      | 0.84|
> | logical flaw       | 60.59|
> | memory          | 4.23|
> | misalign        | 3.38|
> | spatial         | 28.38|
>
> We extend our human-review error analysis to the most and least performant open source models - Qwen2 and LLaVa-v1.6 respectively. Similar to the more powerful closed-source models, the most frequent types of errors involve logical flaws and spatial misunderstandings, at a higher rate than the reviewed closed-source models. The rate of memory flaws, incomplete responses and misalignments is much lower, given that the models exhibit flaws in their logical and spatial reasoning well before the opportunities for other errors to present themselves.
>
> > Weakness 2 & Question 1: The paper notes a performance improvement when diagrams were replaced by textual descriptions, indicating that MLLMs may not be effectively processing visual information. Could the authors provide more qualitative examples or specific insights into how the textual descriptions differed from the diagrams and why these led to improved performance?
>
> From Fig 19 in Pg 44 of the paper,
> Image description:
> Image 1: A rectangle with a diagonal line from the top right corner to the bottom left corner
>
> Image 2: A hexagon with a diagonal line from the top right corner to the opposite corner, dividing the figure into halves.
>
> Image 3: A rectangle with a diagonal line. One end of the line is on the top side, between the top corners; the other end is on the bottom side, between the bottom corners
> Image 4: A hexagon with a diagonal line from the top left corner to the opposite corner, dividing the figure into halves.
>
> Solution:
> Image 3 is the odd one out because it is the only image that does not have a diagonal line that connects opposite corners of the shape. Instead, the diagonal line connects the midpoints of two sides of the rectangle.
>
> Fig 22 in Pg 47 of the paper:
> Image description:
> Image 1: A circle, completely containing 2 discrete circles within it. The discrete circles do not overlap with each other.
>
> Image 2: 2 concentric circles, as well as a separate third circle. The third circle does not overlap at all with the concentric circles.
>
> Image 3: 2 partially overlapping circles, both contained within a single large circle
>
> Image 4: 3 concentric circles.
>
> Let's analyze the relationship between the categories:
> Animals: This is the broadest category, encompassing all living beings.
> Herbivores: A subset of animals that primarily consume plant matter.
> Lions: A specific type of animal that is carnivorous.
> Based on this, we can deduce:
> Lions and Herbivores are distinct subsets of Animals. They do not overlap.
> All Lions and Herbivores are Animals.
> Therefore, the Venn diagram that correctly represents this relationship is Image (1).
> In this diagram:
> The large circle represents Animals.
> The two smaller, non-overlapping circles represent Herbivores and Lions, respectively.
> Both smaller circles are completely contained within the larger circle, signifying that all Herbivores and Lions are Animals.
> This diagram accurately depicts the hierarchical and non-overlapping nature of the three categories.

---

> > ### Author Response · Authors · 2024-11-27
> > **Follow up on the rebuttal**
> >
> > Dear Reviewer,
> >
> > We trust this message finds you well. We are writing to follow up on the rebuttal response we submitted in response to your feedback. We are eager to receive your insights and comments on our response. If there are any areas that you still find unclear or require further clarification, we would be more than happy to provide additional information to ensure a comprehensive understanding of our work.
> >
> > Your feedback is of utmost importance to us, and we greatly appreciate your time and consideration in evaluating our research.
> >
> > Thank you for your attention, and we look forward to hearing from you soon.

---

> > ### Comment · Reviewer_X95T · 2024-12-02
> >
> > Thanks for the detailed response, I'm generally satified with the response and I will increase my rating to 6.

---

### Meta-Review · Area_Chair_7PJ3 · 2024-12-23

**Metareview:**

This paper presents POLYMATH, a benchmark designed to assess the mathematical and visual reasoning capabilities of MLLMs. Reviewers commend the challenging nature of the proposed data and the comprehensive quantitative evaluation conducted on 15 MLLMs with four distinct prompting strategies. The error analysis, revealing that MLLMs rely more on text descriptions than diagrams, is also noted as interesting. However, the main concerns are as follows: 1. The provided analysis lacks novelty, predominantly covering well-established conclusions and thus failing to present novel insights. 2. The experimental analysis is cursory, lacking profound conclusions and omitting in-depth discussions regarding future directions. 3. The paper identifies numerous problems with current LLM models but fails to offer potential solutions. 4. The dataset is considerably imbalanced, which may lead to potentially misleading conclusions. The AC concurs that this benchmarking work could be enhanced by further resolving these issues.

**Additional Comments On Reviewer Discussion:**

The principal concerns put forward by the reviewers are as follows:

1. The provided analysis lacks novelty, predominantly covering well-established conclusions and thus failing to present novel insights.
2. The experimental analysis is cursory, lacking profound conclusions and omitting in-depth discussions regarding future directions.
3. The paper identifies numerous problems with current LLM models but fails to offer potential solutions.
4. The dataset is considerably imbalanced, which may lead to potentially misleading conclusions.

Following the rebuttal discussion, although two reviewers increased their scores to 6, the remaining two reviewers still remain less than fully convinced regarding issues 2, 3, and 4.

I concur that it is of great significance to resolve the aforementioned four issues so as to establish a more valuable benchmark for the community.

---

### Decision · Program_Chairs · 2025-01-22

Reject